# Unveiling diverse coordination-defined electronic structures of reconstructed anatase TiO$_2$(001)-(1 × 4) surface

Xiaochuan Ma [1,2,5], Yongliang Shi [1,5], Zhengwang Cheng [3,5], Xiaofeng Liu[4], Jianyi Liu [1], Ziyang Guo[1,2], Xuefeng Cui[1,2], Xia Sun[1,2], Jin Zhao [1,2], Shijing Tan [1,2] ✉ & Bing Wang [1,2] ✉

Transition metal oxides (TMOs) exhibit fascinating physicochemical properties, which originate from the diverse coordination structures between the transition metal and oxygen atoms. Accurate determination of such structure-property relationships of TMOs requires to correlate structural and electronic properties by capturing the global parameters with high resolution in energy, real, and momentum spaces, but it is still challenging. Herein, we report the determination of characteristic electronic structures from diverse coordination environments on the prototypical anatase-TiO$_2$(001) with (1 × 4) reconstruction, using high-resolution angle-resolved photoemission spectroscopy and scanning tunneling microscopy/atomic force microscopy, in combination with density functional theory calculation. We unveil that the shifted positions of O 2$s$ and 2$p$ levels and the gap-state Ti 3$p$ levels can sensitively characterize the O and Ti coordination environments in the (1 × 4) reconstructed surface, which show distinguishable features from those in bulk. Our findings provide a paradigm to interrogate the intricate reconstruction-relevant properties in many other TMO surfaces.

Transition metal oxides (TMOs) are one of the most important materials due to their versatile physical and chemical properties, such as magnetism[1,2], ferroelectricity[3,4], superconductivity[5,6], and inspire various applications including (photo)catalysis[7,8]. Tremendous research interest has been attracted to interrogate the complex structure-property relationships at the atomic scale, concerning how the coordination between the metal cation and oxygen anion can contribute to the electronic structures[9,10]. The local coordination environments are determined by the coordination number, bond order, length, and angle. However, the coordination environments suddenly change when the bulk structure terminates at a surface. Reconstructions usually take place at the TMOs' surfaces to minimize the surface

energy, accompanied by the rearrangement of electronic structures[11]. The surface region can usually confine 2D/3D electron gas, providing a playground for correlated electron-electron, electron-boson couplings, and spin-charge interconversion[12,13]. Furthermore, lower coordination could appear at the surface defects and step edges, which are recognized as the active center for catalytic reactions[14,15]. These fascinating phenomena ranging from strong correlation to surface catalysis require deep and comprehensive insights into the coordination environments and the corresponding electronic states.

The precise measurement of the coordination environment and electronic property relies on multiply high resolutions in energy, real and momentum spaces. Surface science characterization techniques

[1]Hefei National Research Center for Physical Sciences at the Microscale and New Cornerstone Science Laboratory, University of Science and Technology of China, Hefei, Anhui 230026, China. [2]Hefei National Laboratory, University of Science and Technology of China, Hefei, Anhui 230088, China. [3]School of Science and Hubei Engineering Technology Research Center of Energy Photoelectric Device and System, Hubei University of Technology, Wuhan, Hubei 430068, China. [4]School of Physics, Hefei University of Technology, Hefei, Auhui 230009, China. [5]These authors contributed equally: Xiaochuan Ma, Yongliang Shi, Zhengwang Cheng. ✉e-mail: tansj@ustc.edu.cn; bwang@ustc.edu.cn

have been widely used to measure the geometric and electronic structures of TMOs from different aspects, but each technique has its limitations. In real and momentum spaces, low-energy electron diffraction (LEED) and surface X-ray diffraction (SXRD) can characterize the surface structures[16,17]. Transmission electron microscopy (TEM) can image the cross-sectional atomic arrangements in bulk[18]. Noncontact atomic force microscopy (NC-AFM) with a qPlus sensor[19] has achieved single-bond resolution[20–22]. Scanning tunneling microscopy (STM) has provided a powerful tool in detecting both the surface structures and local density of states[23,24]. However, STM is mainly sensitive to the electronic states in the vicinity of Fermi level ($E_F$) at the near-surface region. The broadly distributed core level and valence band (VB) have usually been examined by X-ray and UV photoemission spectroscopies (XPS/UPS), respectively[25,26]. However, the lack of momentum resolution makes it difficult to identify the band dispersions and thus hard to assign the bands directly. Yet, angle-resolved photoemission spectroscopy (ARPES) has rarely been used to measure the deep valence band electronic structures of TMOs[27,28].

Our study reports that the joint STM-AFM-XPS-UPS-ARPES, in combination with density functional theory (DFT), can provide a comprehensive determination of characteristic electronic structures from diverse coordination environments in the (1 × 4) reconstructed surface of anatase-TiO$_2$(001). By analyzing the measured O $2p$ VB, O $2s$

semi-core level, and Ti $3d$ gap state (GS) and comparing with the calculated electronic structures on the basis of various structural models, the surface electronic states are distinguished from the bulk ones, and able to be correlated to the contributions from the different coordination environments. Such an approach by capturing high-resolution global parameters could be widely applied to other intricated TMO materials.

## Results

### Real-space imaging the surface structures of anatase-TiO$_2$(001)-(1 × 4) using STM and AFM

The bulk anatase-TiO$_2$ consists of 6-fold coordinated Ti atoms and 3-fold coordinated O atoms (O$_{BULK}$). Its terminating (001) surface involves the reduced coordination of 5-fold Ti atoms and the bridging 2-fold O atoms, endowing high surface energy for the well-studied catalytic applications[29]. However, surface science studies employ ultrahigh vacuum annealing that causes the release of (1 × 1) surface energy, leading to the (1 × 4) reconstruction at anatase-TiO$_2$(001) surface[30]. The (1 × 4) superlattice consists of alternated terraces (O$_{TERRACE}$) and hunching-up ridges, where the ridge involves different O rows at the side (O$_{SIDE}$) and the top (O$_{TOP}$) (Fig. 1a). There are two models, namely "ADM"[31] and "AOM"[32], that describe the possible structures of O$_{TOP}$. The main difference is that the ADM ridge involves

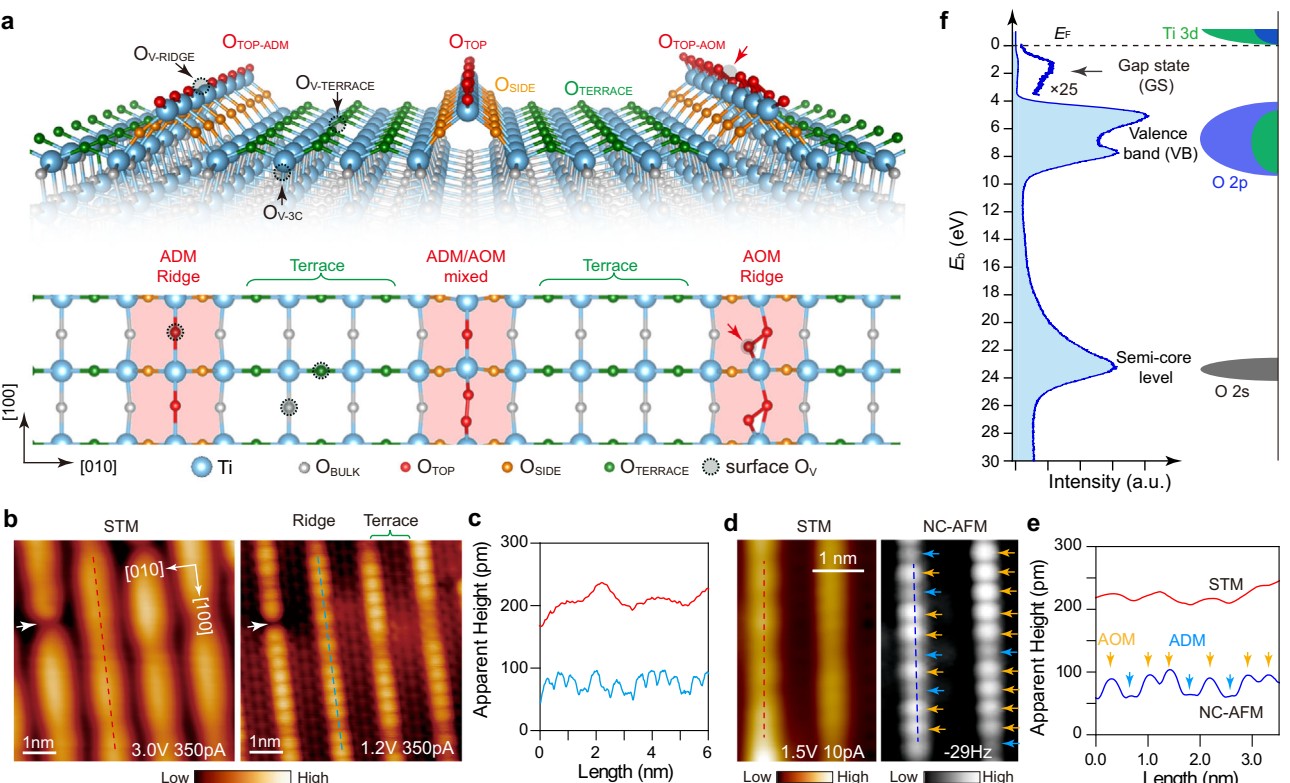

**Fig. 1 | Diverse coordination environments in the (1 × 4) reconstructed surface of anatase-TiO$_2$(001). a** Structural model of the (1 × 4) reconstructed surface. Upper panel: side view, lower panel: top view. The ball-and-stick model sketches three kinds of ridges. Left: the ADM ridge (O$_{TOP-ADM}$), right: the AOM ridge (O$_{TOP-AOM}$), middle: the mixed ADM and AOM ridge with a ratio of 1:1. The O$_{TOP}$, O$_{SIDE}$, O$_{TERRACE}$, and O$_{BULK}$ atoms are correspondingly colored in upper and lower panels. An O$_{TOP-AOM}$ becomes an O$_{TOP-ADM}$ when it is missing an O atom marked by the red arrows. The surface O$_V$ defects at terrace (O$_{V-TERRACE}$ and O$_{V-3C}$) and ridge (O$_{V-RIDGE}$) sites are marked by the black arrows. **b** The empty-state STM images of high bias (3.0 V, 350 pA) and high-resolution (1.2 V, 350 pA) with the same area, measured at 80 K. High-resolution STM image exhibits clear terrace and ridge structures. The white arrow indicates an intrinsic dark point defect[32]. **c** Line profiles extracted from the corresponding colored lines in **b**. The red curve from high bias STM image shows the nonuniform electronic

distributions, and the cyan one from the high-resolved image presents two different heights. **d** A set of in-situ empty-state STM (1.5 V, 10 pA) and the NC-AFM images at a frequency shift of −29 Hz within the same area, measured at 5 K, using a W tip. **e** Line profiles extracted from the corresponding colored lines in **d**. The red curve from STM image shows the nonuniform electronic distributions, and the blue one from AFM presents two distinct contrasts, labeled as ADM (blue arrow) and AOM (yellow arrow), according to the relative height in their models. **f** Left panel: a representative XPS spectrum in the energy range 0–30 eV measured at excitation with light energy of $hv = 200$ eV, sample kept at 20 K, showing the main features of GSs, VBs, and semi-core levels. Right panel: schematic diagram of the main contributions from Ti $3d$, O $2p$, and O $2s$.

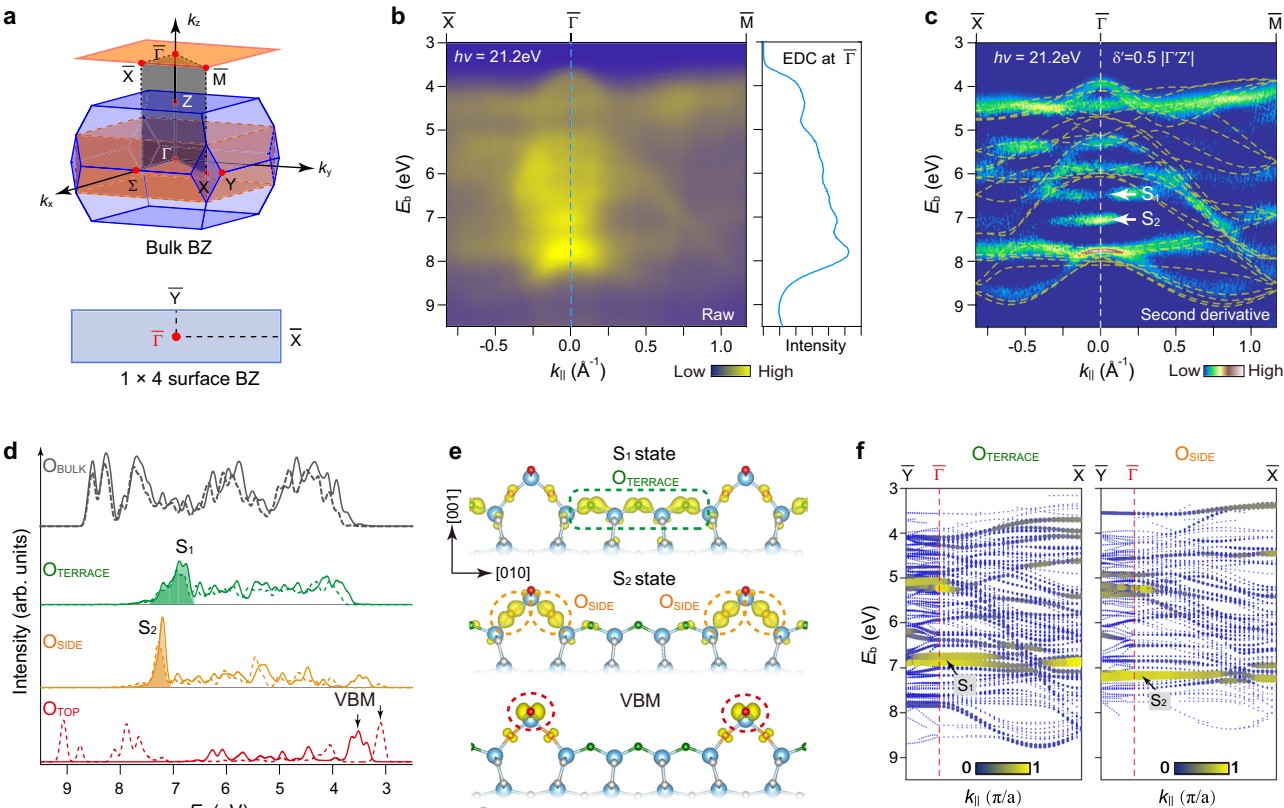

**Fig. 2 | Distinguish the bulk and surface VBs from momentum-resolved O 2p spectra. a** Upper panel: The bulk BZ and the projected surface BZ. Blue shade indicates the bulk *bct* structure of primitive cell. Orange shade indicates the bulk *st* structure of unit cell. The top orange shade is the projected surface 2D structure. Lower panel: surface (1 × 4) reconstructed BZ. **b** The raw ARPES cut excited by $h\nu = 21.2$ eV along the high-symmetry axes and the corresponding EDC at $\bar{\Gamma}$ point. **c** The second derivative cut from **b**, overlaid with the calculated bulk VBs at the corresponding $k_z$ ($\delta' \sim 0.5$) (yellow dashed lines, see details in Supplementary Fig. 1). The calculated VBs have been shifted to maximize the overlap with the experimental results. White arrows: surface levels of $S_1$, $S_2$ bands. **d** The calculated pDOSs of $O_{BLUK}$, $O_{TERRACE}$, $O_{SIDE}$, $O_{TOP\text{-}ADM}$ and $O_{TOP\text{-}AOM}$, based on the ADM (solid lines) and AOM (dotted lines) models. **e** The extracted electronic states of $S_1$, $S_2$ and VBM bands of the ADM model at $\bar{\Gamma}$ point, exhibiting localized spatial distributions at the $O_{TERRACE}$, $O_{SIDE}$, and $O_{TOP\text{-}ADM}$ atoms, respectively. **f** Calculated VB dispersion according to the weightings of surface $O_{TERRACE}$ and $O_{SIDE}$ sites, respectively. The $O_{TERRACE}$ bands involve the main feature of $S_1$ and the $O_{SIDE}$ involve the main feature of $S_2$, respectively.

single $O_{TOP}$ row with 2-fold coordination ($O_{TOP\text{-}ADM}$)[31,33,34], while the AOM ridge involves double $O_{TOP}$ rows with 3-fold coordination ($O_{TOP\text{-}AOM}$)[32,35]. The theory also predicts the possible coexistence of ADM and AOM ridge sites[36,37] (sketched in Fig. 1a). In experiments, the STM images give some signature for their coexistence, for example, the contrast at ridge is quite nonuniform (Fig. 1b, c). However, the STM resolution is not enough to distinguish the ADM and AOM directly[32], remaining a problem to be clarified.

We perform in situ STM and AFM experiments to characterize the structures at a high resolution. The empty-state STM image mainly integrates the Ti electronic state on the ridge and terrace (Fig. 1d, left panel), while the AFM image (given in inverted contrast because of attraction force regime in imaging) mainly reflect the contrasts of O atoms along the ridges (Fig. 1d, right panel). The two groups of O atoms can be distinguished from their relative contrasts, as labeled by ADM and AOM according to the relative heights in their structural models[32,38] (Fig. 1e). Both the STM and the AFM images indicate the coexistence of the ADM and AOM structures on the ridge.

Actually, oxygen vacancy ($O_V$) defects can appear on the surface at either terrace or ridge sites, which act as the reactive centers and introduce metallic states and GSs[39,40]. In Fig. 1a, these coordination environments for O atoms in the anatase-TiO$_2$(001)-(1 × 4) surface are labeled by $O_{BULK}$, $O_{TERRACE}$, $O_{SIDE}$, $O_{TOP\text{-}ADM}$, $O_{TOP\text{-}AOM}$, $O_V$ at ridge ($O_{V\text{-}RIDGE}$), and at terrace ($O_{V\text{-}TERRACE}$ and $O_{V\text{-}3C}$). They together contribute to the complex electronic states, as shown by the measured

XPS spectrum (Fig. 1f), which presents a weak Ti 3d GS between the Fermi level ($E_F$) and VB maximum (VBM), the intense VB mainly consisting of O 2p levels, and a broad peak consisting of O 2s semi-core levels. We now attempt to correlate these electronic states with their coordination environments.

## Distinguishing the surface bands from momentum-resolved O 2p spectra

The anatase-TiO$_2$ has a body-centered-tetragonal (*bct*) structure with the space group $I4_1/amd$[41]. The bulk *bct* Brillouin zone (BZ) of primitive cell (shaded in light blue) is depicted in Fig. 2a. A simple tetragonal (*st*) BZ of unit cell (shaded in orange) is usually applied to examine the band structures along the high symmetric axes[42], which is projected to a surface BZ with (1 × 4) reconstruction along the high symmetric points of $\bar{\Gamma}$, $\bar{M}$ and $\bar{X}$. As known for most TMOs, the VB of anatase-TiO$_2$(001) involves the main contribution from the O 2p states and a minor contribution from the Ti 3d states[43], as shown in the spectrum in Fig. 1f. However, such momentum-integrated spectrum cannot clearly resolve the exact bands. We present the 2D energy-momentum [$E(k_\parallel)$] resolved VB map along the high-symmetry $\bar{\Gamma} - \bar{X}$ and $\bar{\Gamma} - \bar{M}$ directions, with the binding energy ($E_b$) mainly distributing from -3.0 to 9.0 eV below $E_F$ and the energy distribution curve (EDC) at $\bar{\Gamma}$ point (Fig. 2b). Using its second derivative map[44], the contrasts of the band structure in the ARPES map can be highly improved (Fig. 2c). To assign the observed bands, the bulk electronic bands have been calculated by

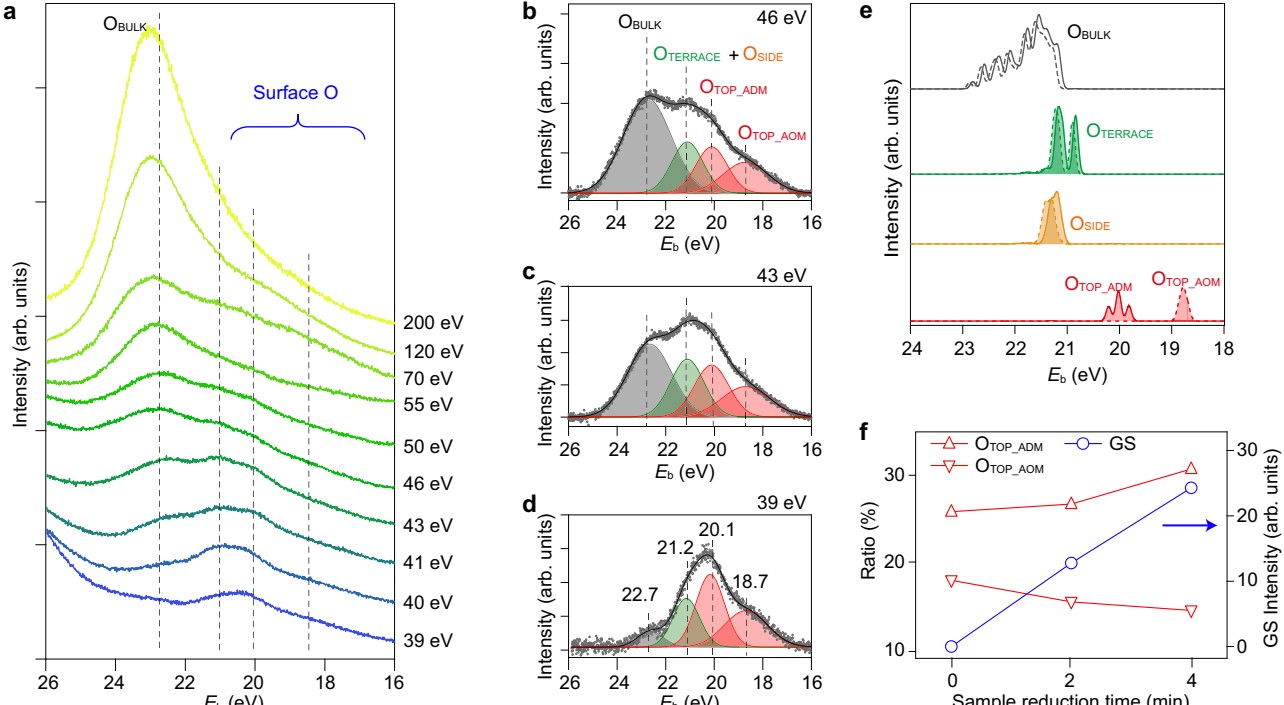

**Fig. 3 | Recognizing the surface core levels from O 2s spectra. a** Evolution of O $2s$ semi-core level spectra with increasing excitation photon energy from 39 eV to 200 eV. **b–d** Peak fitting of O $2s$ spectra with BG subtracted for 39, 43, and 46 eV excitation, respectively. Four peaks at 22.7, 21.2, 20.1, and 18.7 eV can be extracted with relative intensity changes. **e** The calculated O $2s$ semi-core level pDOSs of each O atom. **f** Plot of the $O_{TOP-ADM}$ (red triangle) and $O_{TOP-AOM}$ (red inverted triangle) contributed peak area ratio in the left axis and the corresponding Ti $3d$ GS intensity (blue circle) in the right axis as a function of sample reduction. The as-grown sample was gradually reduced by increasing Ar⁺ sputtering (each cycle 2 min) and annealing at 650 °C.

using both the primitive cell and the unit cell[41,42], as well as different $k_z$ (see calculation details in Supplementary Fig. 1). The calculated bulk VBs can overall reproduce the experimental results, except for the two at $E_b \sim 6.5$ eV and 7.1 eV, as labeled by $S_1$ and $S_2$, respectively (Fig. 2c).

We turn to consider the $S_1$ and $S_2$ as the VBs contributed by surface O atoms. To verify their surface nature, we carried out the experiment by depositing potassium (K) atoms to the surface because the alkali atoms can strongly modify the surface levels but have little influence on the bulk bands. It is clear that the $S_1$ and $S_2$ can be quenched by K atoms adsorption and recovered after K desorption, similar to behaviors of the $H_2$ adsorption on Ir surface[45], while the bulk bands remain unchanged (Supplementary Fig. 2). Different from the bulk VBs with broadly distributed $E(k_{||})$ dispersions, the $S_1$ and $S_2$ are nearly flat near $\bar{\Gamma}$ point. Such unique features imply the $S_1$ and $S_2$ may be contributed by certain specific O sites in the surface. We calculated the surface electronic structures using the slab models by including the geometries of ADM and AOM models, respectively. The projected density of states (pDOSs) of each O atoms of $O_{TERRACE}$, $O_{SIDE}$, $O_{TOP-ADM}$ and $O_{TOP-AOM}$ are extracted to distinguish their contributions (Fig. 2d). For $O_{TERRACE}$ and $O_{SIDE}$ atoms, it is found that both ADM (solid lines) and AOM (dotted lines) models give approximate electronic structures within a broadly hybridized energy range of $E_b \sim 3$–8 eV. But, the $O_{TERRACE}$ atoms dominate at $E_b \sim 6.8$ eV (green shaded) and the $O_{SIDE}$ atoms dominate at $E_b \sim 7.2$ eV (orange shaded) with well-defined spatial distributions (Fig. 2e), which could be readily assigned to the observed $S_1$ and $S_2$ bands, respectively. Furthermore, we plot the band dispersion of each O atoms calculated using the ADM model (Supplementary Fig. 3). The weightings of surface O atoms show localized features relative to the that of bulk O atoms. The weightings of $S_1$ and $S_2$ indicate near flat character in the vicinity of $\bar{\Gamma}$ point (Fig. 2f), in good agreement with those observed in the experimental $E(k_{||})$ map. For $O_{TOP}$ atoms, it is known that they mainly contribute to the VBM[46,47] with

localized features (Fig. 2e). The calculated pDOS of $O_{TOP-ADM}$ and $O_{TOP-AOM}$ with ADM and AOM models, respectively, dominate at VBM, but with slight energy difference (Fig. 2d). From the O $2p$ spectra, it is still difficult to distinguish $O_{TOP-ADM}$ and $O_{TOP-AOM}$, we thus search for more evidence from the O $2s$ spectra in the following sections.

**Recognizing the surface semi-core levels from O 2s spectra**
The O $1s$ XPS spectrum of anatase-$TiO_2$(001)-(1 × 4) exhibits only a single peak, as widely observed in the crystalline $TiO_2$ surfaces[48,49]. By contrast, even under excitation by a high photon energy (610 eV), the O $2s$ spectrum of anatase-$TiO_2$(001)-(1 × 4) presents a tail spreading to a low $E_b$ region, where the tail could be from the contributions of the surface O atoms (Supplementary Fig. 4). Distinctly different features can be observed in comparison with the results from a simpler rutile-$TiO_2$(110)-(1 × 1) surface at excitations using lower photon energies, which can be more sensitive to the surface states (Supplementary Fig. 5a, b). By tuning the excitation photon energies ($h\nu$) from 39 eV to 200 eV, the evolution of these peaks can be more clearly shown (Fig. 3a). According to the universal curve for the electron inelastic mean free path (IMFP) as a function of the kinetic energy[50], it can be expected that the spectra excited with $h\nu > 100$ eV could mainly detect the bulk information, while those excited with $h\nu \sim 30$–100 eV are sensitive to surface region. The spectra excited with $h\nu = 120$ and 200 eV show a strong peak at $E_b \sim 22.7$ eV, which could be assigned to the contributions of $O_{BULK}$ atoms (Fig. 3a), similar to the results from the rutile $TiO_2$(110) surface[51]. As the $h\nu$ reduced to below 100 eV, the intensity of the peak at $E_b \sim 22.7$ eV decreased dramatically, and meanwhile, several new peaks at lower $E_b$ arise. Three selected spectra with $h\nu = 39$, 43 and 46 eV excitation are presented in Fig. 3b–d, which can be decomposed into four peaks by the best XPS fitting (Supplementary Fig. 6). Except for the peak at $E_b \sim 22.7$ eV that is contributed by $O_{BULK}$ atoms (gray shaded), the three peaks at $E_b \sim 21.2$, 20.1, and

18.7 eV could be contributed by surface O atoms, as a result of the chemical shifts[52] of the surface semi-core levels. It is noted the intensity of these distinguishable peaks are changing dramatically: for the one at $E_b$ ~ 22.7 eV of $O_{BULK}$ atoms, the enhanced intensity could be reasonably attributed to the increased IMFP with larger $h\upsilon$; for the other three peaks of surface O atoms, we need to consider some other possible factors, like the photoelectron diffraction and the resonant photoemission processes. We can exclude the effect of the photoelectron diffraction due to the possible diffraction-caused enhanced angle-dependent intensity variations (Supplementary Fig. 7). The angle distribution curves (ADCs) at each photon energy give overall Gaussian-like shape for the integrated intensity distributions against $\theta$, obviously no diffraction-enhanced intensity with ±15°. The EDCs show nearly the same feature at each excitation photon energy, showing the angle-independent O $2s$ spectra. These analyses can exclude the possibility of the diffraction effect by the reconstructed surface structure. While, by measuring the spectra using the tunable excitation phonon energy in the range of $h\upsilon$ ~ 39–55 eV, the resonant photoemission processes were observed to occur at around $h\upsilon$ ~ 43–46 eV (Supplementary Fig. 5c), which could be assigned to the Ti $3p \rightarrow 3d$ optical transition at anatase-TiO$_2$(001) surface[53,54]. Such resonant photoemission processes could enhance the photoemission intensities of the surface semi-core levels at certain excitation photon energy, and make the peaks more distinguishable in the O $2s$ spectra. Nevertheless, this effect does not contribute any additional peak and much easily be recognized according to the analysis of our results.

To understand these O $2s$ semi-core level peaks, we analyzed the calculated pDOS in Fig. 3e. The pDOS of $O_{BULK}$ has a broad spectral weight in the range $E_b$ ~ 21–23 eV, which corresponds well to the peak of $E_b$ ~ 22.7 eV in the experiment. The pDOS of $O_{SIDE}$ and $O_{TERRACE}$ sites overlap in the energy range of $E_b$ ~ 20.8–21.4 eV (Fig. 3e). These two together could be probably correlated to the peak of $E_b$ ~ 21.2 eV in the experiment. More interestingly, the pDOS of $O_{TOP-ADM}$ and $O_{TOP-AOM}$ atoms are clearly separated at $E_b$ ~ 20.1 eV and 18.7 eV, respectively, which are in excellent agreement with the two lower energy peaks in the experiment (red shaded in Fig. 3b–d). Based on the above assignments, the existence of the two peaks at $E_b$ ~ 20.1 eV and 18.7 eV implies the coexistence of ADM and AOM ridge structures at the anatase-TiO$_2$(001)-(1×4) surface, in line with the two distinct contrast of ADM and AOM in the AFM measurements (Fig. 1d).

The quantitative analysis of the mixed ADM and AOM ratio is difficult, because the absolute areas of the two peaks are sensitive to the excitation $h\upsilon$ and rely on how the background signal are subtracted. Qualitatively, we investigated the change of ADM and AOM ratio at the excitation $h\upsilon = 40.8$ eV using slightly reduced surface through treatments of sputtering and annealing cycles (Supplementary Fig. 8). The reduction procedure is monitored by the increase of Ti $3d$ GS intensities (blue circle) in Fig. 3f. It can be seen that the $O_{TOP-ADM}$ component (red triangle) is increasing while the $O_{TOP-AOM}$ component (red inverted triangle) is decreasing during the sample reduction procedure, which strongly supports our assignments. To further examine the stability of mixed ADM-AOM structure of anatase TiO$_2$(001)-(1×4), we calculate the surface energy correlation with the oxygen chemical potential and the phase diagram (Supplementary Fig. 9), in consistent with the similar phase diagrams for AOM and ADM in previous stusies[47,55,56]. Our study provides additional information for the mixed ADM-AOM structures with tunable ratios. The results indicate that the mixed ADM-AOM structure could be a stable phase during sample growth under a relatively low temperature and pressure conditions.

### Identifying the O defect states from Ti $3p$ and $3d$ spectra

The O $2p$ and $2s$ spectra above have successfully identified the different coordination environments of O atom at the reconstructed anatase-TiO$_2$(001)-(1×4) surface. In addition, the $O_V$ defects commonly exist on oxides' surface causing excess electron doping at the surface. The electron doping could introduce two features: (i) a delocalized electron gas at near-surface region confined by the band bending potential, which forms a metallic state (MS) just below $E_F$ (refs. 39,40); (ii) localized electron denotation to nearby Ti atoms, which reduces Ti$^{4+}$ to Ti$^{3+}$ and forms Ti $3d$ GSs[32,39,40]. The $O_V$ defects could be produced by either sputtering and annealing in vacuum[32], or light irradiation with a high flux radiation beam[39]. We here present the evolution of Ti $3p$ and $3d$ spectra with $O_V$ formation under light irradiation. By increasing the light irradiation time, the intensity of Ti$^{4+}$ state gradually decreases while the intensity of Ti$^{3+}$ state arises in the Ti $3p$ spectra (Fig. 4a). Meanwhile, both MS and GSs arise in the Ti $3d$ spectra (Fig. 4b). The intensities of GSs increase gradually, while that of MS looks rapidly saturating at $E_F$. This is because the MS is formed due to the lowered Ti $3d$ band below $E_F$, which has a saturated electron density that leads to the saturated intensity near the $E_F$ (ref. 39). Two distinct GSs appear at $E_b$ ~ 1.1 and 1.8 eV in the Ti $3d$ spectra, as denoted by GS1 and GS2, respectively (Fig. 4b). Different from the rutile-TiO$_2$(110)−1×1 surface, where the $O_V$ defects can only produce one GS at $E_b$ ~ 0.9 eV[57], the emergence of two GSs on anatase-TiO$_2$(001)-(1×4) surface is unique, which had been observed in previous studies[39,40] but have not been understood yet. Bigi et al. found the adsorption of O$_2$ at defective anatase-TiO$_2$(001)-(1×4) surface can quench both the GS1 and GS2, and thus demonstrate their origin should be from surface defects rather than from bulk defects[40]. We extract the intensities of the two GSs separately from the best fitting (Supplementary Fig. 10). It is found that although the intensities of Ti$^{3+}$ state (Fig. 4c) and the total GSs (Fig. 4d) are increasing meanwhile as a function of light irradiation time, the intensities of GS1 and GS2 are changing asynchronously (Fig. 4d), implying the GS1 and GS2 must have different origins from surface defects. In our previous experiments, we have detected the point defect induced GS at $E_b$ ~ 0.9 eV at ridge sites by scanning tunneling spectroscopy[32], and the hydroxyl groups induced GS at $E_b$ ~ 1.6 eV at terrace sites by UPS[49]. Considering the approximated energies, we assign the observed GS1 at $E_b$ ~ 1.1 eV and GS2 at $E_b$ ~ 1.8 eV to the electronics states of $O_V$ defects at ridge and terrace sites, respectively.

We calculate the configurations and formation energies for possible surface $O_V$ defects at ridge and terrace sites using spin-polarized DFT (Supplementary Fig. 11). From the formation energies, it is found that the missing of one $O_{TOP}$ is the most possible $O_V$ defect at the ridge ($O_{V-RIDGE}$, Fig. 4e), and the missing of a bridging O atom along [100] direction or along [010] direction could be the $O_V$ defect at terrace ($O_{V-3C}$ and $O_{V-TERRACE}$, Fig. 4f). The calculated electronic structures with PBE + U functional (U = 3.9 eV)[40] show that each $O_V$ defect can contribute to a GS with its excess electron redistributing to adjacent Ti atoms (Fig. 4e, f). Further ab initio molecular dynamics (AIMD) simulations indicate the excess electrons are stable at the adjacent Ti atoms (Supplementary Fig. 12). The charge redistribution and the distortion of the lattice in the vicinity imply the formation of small electron polaron, similar as the small polaron of $O_V$ at rutile-TiO$_2$(110) surface[58]. But, from the pDOS with either antiferromagnetic[59] or ferromagnetic states (Supplementary Fig. 13), the energies of different $O_V$ defects are not separated clearly, making the assignment difficult. Such energy inaccuracy is possibly because of the self-interaction error in DFT, which usually leads to an underestimated bandgap of TiO$_2$. In particular, when multiple $O_V$ defects are considered (Supplementary Fig. 14), the GSs show very close energies within the underestimated bandgap of 2.5 eV (Fig. 4g).

It is difficult to create $O_V$ defects at either ridge or terrace in a controllable manner for verifying the assignment of GS1 and GS2. Alternatively, we design an experiment to create hydroxyl groups at ridge and terrace sites from methanol (CH$_3$OH) dissociation, by considering the hydroxyl groups can provide surface states similar to those of the $O_V$ defects owing to the induced excess electrons that

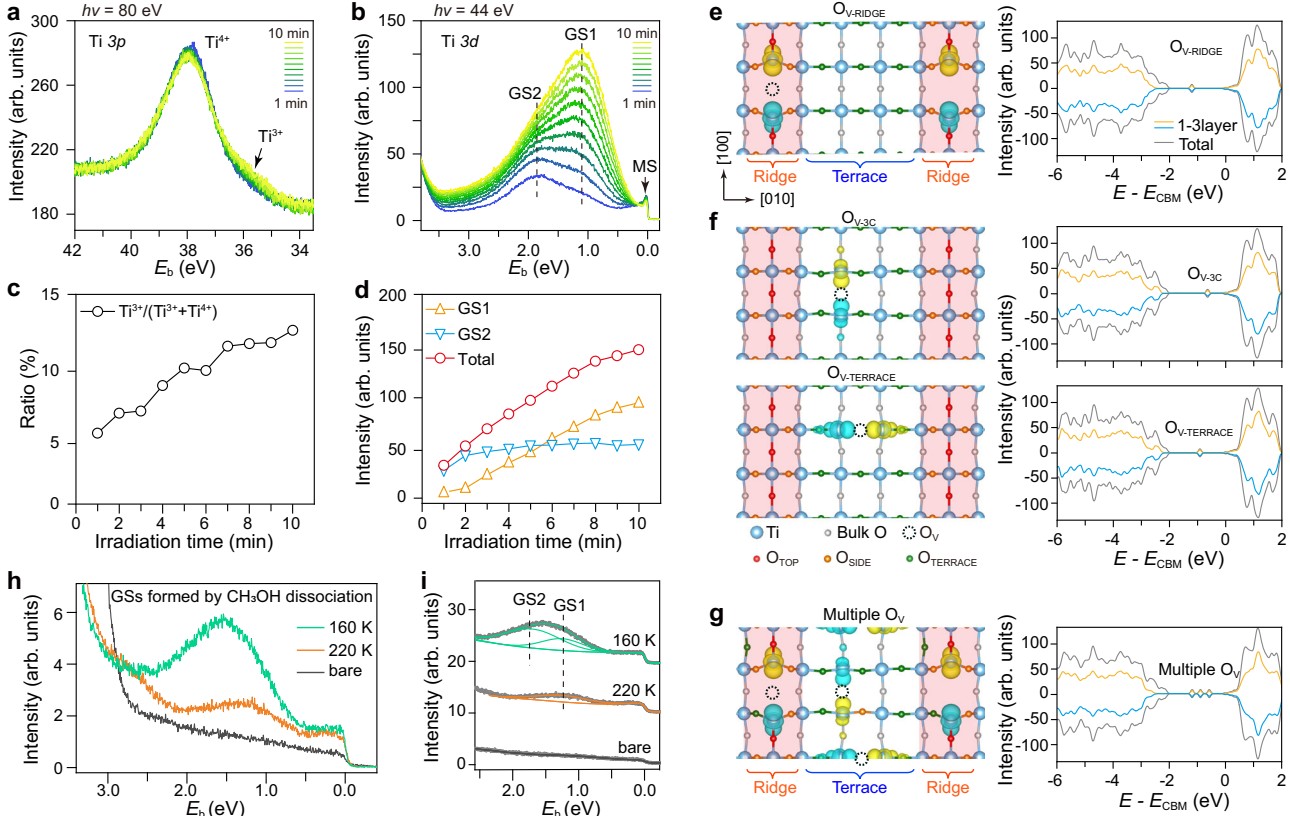

**Fig. 4 | Recognition of different $O_V$ defects induced Ti $3d$ GSs. a, b** Ti $3p$ core level spectra obtained by $hv = 80$ eV and the in-gap UPS spectra excited by $hv = 44$ eV (synchrotron radiation light), respectively. **c, d** Evolution of the $Ti^{3+}/(Ti^{3+}+Ti^{4+})$ ratio, and the intensities of GS1, GS2 and total of (GS1 + GS2), as a function of light irradiation time. **e–g** Calculated charge density contours of excess electrons and DOSs of total (gray) and 1–3 layer (yellow: spin up, blue: spin down) with an $O_{V-RIDGE}$ at ridge **e**, an $O_{V-3C}$ or $O_{V-TERRACE}$ at terrace **f** and multiple Ov defects at ridge and terrace **g**, respectively. **h, i** The in-gap UPS spectra obtained after 30 min UV irradiation (He lamp, 21.2 eV) with $CH_3OH$ adsorption at 160 and 220 K, respectively. In **i**, the UPS spectra are fitted after background subtraction with one peak at 1.20 eV for the spectrum at 220 K, and double peaks at 1.20 and 1.75 eV for the one at 160 K. A spectrum measured at the bare surface is used for reference (gray).

contribute to the GSs. Methanol has similar adsorption behavior as water[49], that the molecular $CH_3OH$ only adsorbs at ridge sites at $T > 190$ K, but adsorbs at both ridge and terrace sites at $T < 190$ K. Therefore, at $T > 190$ K, a single GS is expected to appear for ridge sites after $CH_3OH$ dissociation, while at $T < 190$ K, double GSs for ridge and terrace sites are observed. Figure 4h compares the spectra obtained after $CH_3OH$ dissociation with samples prepared at 160 K and 220 K, respectively. Clearly, for the spectrum at 220 K, only one GS peak at ~1.20 eV can be detected; for the spectrum at 160 K, the peak is broadened to higher energies. The best fit can distinguish two peaks at about $E_b$ ~ 1.20 eV and 1.75 eV (Fig. 4i), which could be assigned to the hydroxyl groups induced GSs at ridge and terrace sites, respectively. Such similarity strongly suggests that the excess electron doping by either $O_V$ defects or hydroxyl groups can form a GS at $E_b$ ~ 1.1–1.3 eV at the ridge and a GS at $E_b$ ~ 1.6–1.8 eV at the terrace.

## Discussion

The above measurements unveil the surface electronic states in the intricated reconstructed surface of anatase-$TiO_2$(001), which presents diverse O coordination environments. Using the multi-technique approach, we distinguish the bands of $O_{TERRACE}$ and $O_{SIDE}$ using the energy-momentum-resolved O $2p$ spectra, the electronic states of $O_{BULK}$, $O_{TOP-ADM}$ and $O_{TOP-AOM}$ using the O $2s$ spectra, the defect electronic states at ridge and terrace sites through measuring the gap-state Ti $3d$ spectra. These global parameters together establish the correlation of the surface electronic states with the (1 × 4) reconstructed structure of anatase-$TiO_2$(001), and provide the deeper understanding of the mixed ADM-AOM configurations[36,37]. The revealed coordination

(structure)-electronic (property) relationships could pave the way to understand the widely concerned surface catalysis and correlation phenomena of anatase-$TiO_2$. For example, the main focus in photocatalytic water splitting on $TiO_2$ is to understand the energy level alignment between the $TiO_2$ VBs and the highest occupied molecular orbitals (HOMOs) of water at their interface[60], which requires an insight into the intricate coordination environments and surface electronic structures. Distinguishing the surface VBs of $O_{TOP}$, $O_{TERRACE}$ and $O_{SIDE}$ can directly locate and determine how the water HOMOs can hybridize with the $TiO_2$ VBs with energy-momentum-site specificities; regarding the correlation phenomena, the anatase-$TiO_2$ can provide a platform to tune the electron-phonon and electron-plasmon couplings through the excess charge by $O_V$ defect[39,61] and photo-excitation[62,63], and can support an enhanced electron-phonon coupling interface of FeSe/anatase-$TiO_2$ to induce high-$T_c$ superconductivity[64]. Distinguishing the origins of the doped electrons from different GSs can provide prerequisites to tune the electron-boson couplings in the complicate many-body interactions. In summary, our results provide the comprehensive understanding of the complex coordination and the corresponding electronic structures of the prototypical anatase-$TiO_2$(001), and beyond, may benefit to other TMO materials.

## Methods

### Sample preparation

The anatase-$TiO_2$(001) thin films were epitaxially grown on 0.7 wt% Nb-doped $SrTiO_3$(001) substrates by pulse laser deposition (PLD) method[39]. During deposition, The $O_2$ pressure was kept at $1.5 \times 10^{-3}$ Pa and the substrate temperatures were kept at a 650 °C. To obtain high

quality 1 × 4 reconstructed surface, the anatase-TiO$_2$ films were grown slowly on SrTiO$_3$ substrate with deposition rate of ~1 nm/h. The typical thickness of the thin films was 20–30 nm. The as-grown anatase-TiO$_2$(001) samples were transferred between the PLD, STM and ARPES systems without exposure to air, via a portable ultrahigh vacuum transfer chamber with a battery powered ion pump to maintain a pressure better than 1 × 10$^{-9}$ mbar.

### STM, XPS, and ARPES measurements

The STM measurements were conducted in a low temperature STM (Omicron LT, Matrix) at 80 K with a tungsten tip. High-resolved in situ STM and AFM experiments with a qPlus sensor were conducted at 5 K. The photoelectrons in XPS and ARPES measurements at 90 K were excited by the Mg Kα radiation (1253.6 eV) and the resonance He lamp (21.2 eV and 40.8 eV), respectively. The $h\nu$-dependent XPS measurements were performed at the beam line 09U of Shanghai Synchrotron Radiation Facility (SSRF), with the photon energy varying from 39 to 610 eV with horizontally polarized light at 20 K. Hemispherical energy analyzer (VG Scienta, DA30L) was used for all measurements. Energy resolution of instrument was better than 5 meV and the angular resolution was better than 0.1°.

### DFT calculation

All the calculations are performed with the Vienna Ab-initio Simulation Package (VASP). Electron-nuclei interactions are described by the projector-augmented wave pseudopotentials[65]. The Perdew-Burke-Ernzerhof (PBE) exchange-correlation functions are employed in all calculation[66]. The long-range van der Vaals interactions are corrected with Grimme's D2 method[67]. The Kohn-Sham wave functions are expanded in plane waves up to 500 eV. The structures are relaxed until the atomic forces are less than 0.01 eV/atom and total energies are converged to 10$^{-5}$ eV. A 15 Å thick area of vacuum in the z-direction and dipole correction are employed to avoid non-physical interaction between neighboring slabs.

### Band calculation.

The 3D BZs are sampled with a grid of 13 × 13 × 13 k-points according to the Monkhorst-Pack procedure for primitive cell and unit-cell of bulk anatase-TiO$_2$, respectively[68]. The anatase-TiO$_2$(001)-(1 × 4) surface was modelled using a slab of four O-Ti-O trilayers based on ADM and AOM model, with the bottom O-Ti-O trilayer being fixed and 3 × 9 × 1 k-points are used for this surface. To assign the three new peaks in detail, we calculate the electronic structures using DFT with the hybrid HSE06 functional.

### Surface energy and phase diagram calculation.

Using the (4 × 4) supercell, we investigate different oxidized surfaces for ADM: AOM corresponding to ADM (not oxidation), 3:1 (25% oxidation), 1:1 (50% oxidation), 1:3 (75% oxidation) and AOM (all oxidation). The stability of these structures in different environments can be discussed by calculating the free energy correlation with the oxygen chemical potential. The chemical potentials must satisfy the following boundary conditions: (i) $\mu_O \leq \frac{1}{2}E_{O_2}$; (ii) $\mu_{Ti} \leq \mu_{Ti}^{bulk}$ and (iii) $\mu_{Ti} + 2\mu_O = E_{TiO_2}$, in which $E_{TiO_2}$ is the internal energy of the bulk TiO$_2$ per unit cell. The stability of surface energy can thus be written as:

$$\gamma = \frac{1}{N}(E_{tot} - E_{ref} - n_{Ti}\mu_{Ti} - \mu_O n_O) \tag{1}$$

According to the constraints of the chemical potential, the range of $\mu_O$ is set as $\mu_{O_2}/2 \geq \mu_O \geq (E_{TiO_2} - \mu_{Ti})/2$. In equilibrium with O$_2$ gas, $\mu_O$ is expressed as:

$$\mu_O = \frac{1}{2}[E_{O_2} + \Delta H_{O_2}(T,P^0) - T\Delta S_{O_2}(T,P^0) + k_B T\ln(P/P^0)] \tag{2}$$

where $k_B$, $P^0$ and $P$ are the Boltzmann constant, standard atmospheric pressure, and oxygen partial pressure, respectively. $E_{O_2}$ is the total energy of the O$_2$ obtained from a spin-polarized calculation. $\Delta H_{O_2}(T,P^0)$ and $T\Delta S_{O_2}(T,P^0)$ are taken from a thermodynamic database[69]. The last term is the contribution coming from the partial pressure of oxygen.

### O$_V$ defects calculation.

The geometry optimizations and electronic states of O$_V$ defects at the ridge and terrace sites are simulated in 4 × 3 supercell, which supplies enough room to distort the lattice for electron. We perform the spin-polarized DFT calculation using the PBE method to calculate the configurations and formation energies of possible surface O$_V$ configurations. Due to the self-interaction error[70], the orbitals are too delocalized in DFT calculation to locate an electron to an individual Ti atom. To overcome this problem, we perform the spin-polarized DFT calculation using the PBE + U method with U = 3.9 eV[40] to calculate pDOS and electronic charge distribution. After geometry optimization, we use velocity rescaling to bring the temperature of the system to 300 K. An 8 ps microcanonical ab initio molecular dynamics (AIMD) trajectory is generated.

## Data availability

All data needed to support the conclusions in the study are available within the article and/or the supplementary files. Data underlying Figs. 1–4 and Supplementary Figs. 1–14 are provided in the source data file with this paper. Source data are provided in this paper.

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

## Acknowledgements

This work is supported by the Innovation Program for Quantum Science and Technology (2021ZD0303302), the CAS Project for Young Scientists in Basic Research (YSBR-054) and the CAS Strategic Priority Research Program (XDB36020200), National Natural Science Foundation of China (21972129, 11904349), Anhui Initiative in Quantum Information Technologies (AHY090300), and the New Cornerstone Science Foundation. X.M. acknowledges the support of the National Natural Science Foundation of China (22302188). X.S. acknowledges the support of the National Natural Science Foundation of China (12074361).

## Author contributions

B.W. and S.T. designed and supervised the research; J.Z. and X.S. supervised the calculations; X.M. contributed to the sample preparation. X.M., Z.C. and Z.G. performed the UPS, XPS, and ARPES experiments; Y.S. and X.L. performed the theoretical calculations; J.L. and X.C. performed the STM and AFM experiments; X.M., X.C., S.T. and B.W. contributed to the data analysis. S.T. and X.M. generated the figures and wrote the manuscript; All authors discussed the results and commented on the manuscript at all stages.

## Competing interests

The authors declare no competing interests.
