## [Peer Review File · Nature Communications]

Unveiling diverse coordination-defined electronic structures of reconstructed anatase TiO₂(001)-(1×4) surfaceReviewers' comments:

Reviewer #1 (Remarks to the Author):

This interesting work reports the determination of distinct coordination environments of the prototypical binary oxide anatase-TiO₂(001)-(1×4). The integration of momentum, energy, and real-space resolutions in O2p, O2s, and Ti3d spectra, coupled with scanning tunneling microscopy and density functional theory, constitutes a significant advancement in surface science and catalysis. However, the reliability of this innovative approach necessitates further validation using reference systems, preferably simpler ones with fewer complex coordination environments. The current limitations of the approach are evident in the results and conclusions, highlighting the need for broader applicability. Additionally, several issues must be addressed before considering the publication of this work:

1. Inferring the existence of ADM and AOM configurations solely from the STM image is initially imprecise. STM image contrasts can stem from various sources, and deducing the presence of ADM and AOM solely based on one image is not entirely convincing. While the subsequent sections of the article aim to validate the existence of these structures through spectroscopy, robust experimental evidence remains incomplete.
2. For the 1:1 mixed ADM-AOM structure, there might be another setup where two AOM patterns are together. Can its energy related to the oxygen chemical potential be added to Figure S6? Also, it would help if the formula used to calculate the phase diagrams for the different ADM-AOM structures is provided.
3. Given the many ways Ti³⁺ can be arranged near the Ov site, it's essential to look at all these configurations to determine the most stable configuration of the defective anatase-TiO₂(001). It's not clear if the authors used spin-polarized calculations during the structural relaxation processes for the defective anatase-TiO₂(001) or if it was only used for single point electronic structure calculations.
4. The authors have given theoretical proof for the presence of both ADM and AOM structures. But is there direct proof for both defective structures being present at the same time? What are the formation energies for these two kinds of oxygen vacancies?
5. The study seems to focus on single Ov. What happens when there are multiple Ov, in terms of shape and electric properties? How do these findings compare to what's seen in experiments?

Reviewer #2 (Remarks to the Author):

This is an interesting paper, although I do not consider it as a major advance. The authors certainly oversell the work. There is also a major need for extensive correction of the grammar and English which is quite bad in many places.

I have a few comments:

Abstract needs work: The sentence "We resolve each..." needs rewriting.

The term "multi-domain" seems to be wrong.

Many other parts of the text need checking by a native English speaker, they are somewhat clumsy or have the wrong words. I won't comment further.

Please replace reference 21 by an original reference, it is a recent derivative. The method has been known for 40 years.

The O2s is semicore, not core. Was relaxation or a Slater $\frac{1}{2}$ method used? It is standard that positions and even shapes are not correct if just the pDOS is used.

Reviewer #3 (Remarks to the Author):

The authors report a multi-technique study (STM, UPS, XPS, ARPES, DFT) to characterize the (4x1) reconstructed TiO₂ anatase (001) surface. This is certainly an interesting and important surface, and new knowledge about it is always a good thing.

A model for the (4x1) surface was described by Lazzeri and Selloni (ref. 30) and later a somewhat modified version by Wang et al (ref. 31).

The authors analyze differentiated ARPES valence band spectra and, with the help of DFT calculations and projected DOS, assign them to different O atoms.

The authors argue that they are able to distinguish the signals in the valence band for O atoms in 7 different environments. This is a rather naïve view of electronic structure, to the extent that I would view this analysis as an over-interpretation. All bands are hybridized. It makes not much sense to assign them to individual O atoms.

The paper then goes on to analyze O2s (shallow core levels), taken at different photon energies, and, again assign them to different O atoms according to PDOS calculations. This makes a little bit more sense, but I am doubtful that the rather pronounced changes in the core levels would only be due to IMFP length effects, as claimed. Then this should lead to a rather smooth change, which should not be very pronounced in the hv range considered (39 to 200 eV).

The observed relatively drastic intensity changes could be due to two phenomena: 1) hybridization with Ti band (and the well-known resonant photoemission effect) and/or 2) if taken with the energy analyzer set to a narrow angular acceptance range, photoelectron diffraction.

As far as I can see, neither of these effects has been considered.

The analysis of the Ti3p core level spectra and the assignment of the defect state to different O vacancies makes sense only if the excess electrons are in a small polaron state right next to the vacancy. This is possibly true, but more work would be needed to show this.

Summarizing, the interpretation of the experimental data is naïve at best, and completely wrong at worst.

I suggest rejecting this paper.

Reviewer #4 (Remarks to the Author):

The paper by Ma et al. reports on the assignment of different coordination structures for anatase TiO₂(001) with their corresponding electronic structures. The work is a multi-technique experimental approach combined with DFT calculations. While STM images provide information about the coexistence of two specific coordination environments at the ridge sites, with ARPES and XPS, the authors get insights into the electronic structure. However, it is the combination of these experimental techniques with theoretical DFT calculations that allows the determination of seven local coordination environments in the anatase TiO₂(001)-(1x4) surface reconstruction. The subject of this research is of great interest due to the relevance of this material, TiO₂, in different the technological applications, being of special importance in catalysis due to its surface reactivity. The manuscript achieves a high enough scientific

ranking to be accepted in Nature Communication. The authors' experiments are well done and well thought out and theoretical calculations are fundamental in assigning the different electronic states to the coordination environments. The work not only demonstrates the different coordination structures of the anatase TiO₂(001)-1x4 surface, but also provides a paradigm to explore the structure and electronic properties of TMOs. However, there are some minor point that the authors need to address:

- Catalysis and photocatalysis are not intrinsic properties of materials per se, but rather phenomena that arise from the interaction between materials and chemical reactions (and light in the case of photocatalysis). Therefore, it is inaccurate for the author to state that TMOs "exhibit versatile functional properties such as catalysis and photocatalysis" being more appropriate that TMOs possess versatile functional properties suitable for various applications, including catalysis and photocatalysis. Similarly, strong correlation is not a property but a phenomenon. The sentence "These fascinating properties ranging from strong correlation to surface catalysis ..." is not entirely correct. More precisely, the authors may refer to "These fascinating phenomena ranging from strong correlation to surface catalysis ...".

- In page 7 and 8, "OTOP-ADM and OTOP-ADM" is written several times instead of "OTOP-ADM and OTOP-AOM".

- In the case of the O-2s spectra fits (Fig. 3b-c), are the minimum four subspectra needed to match the XPS O-2s curve? Would it be possible to achieve a good fit with fewer curves? Clarify in the manuscript.

- The metallic state in the XPS Ti-3d spectra (Fig. 4) appears already after 1 min of light irradiation and remains almost constant for longer. Comment on it.

- In this work, the assignment of the two gap states in the Ti-3d XPS spectra, which were previously observed but not understood until now, is quite important. However, the Ovrige GS has two peaks and, although they are centered at 1.1 eV, one is located at 1.6 eV. This BE value is the same as the GS2 contribution. Could it interfere with the GS assignment? Add some discussion in the manuscript to clarify this point.

- Several sentences in the manuscript suffer from clarity and grammar issues. The manuscript requires improvement to achieve linguistic correction.

Reviewers' comments:

Reviewer #1 (Remarks to the Author):

This interesting work reports the determination of distinct coordination environments of the prototypical binary oxide anatase-TiO₂(001)-(1×4). The integration of momentum, energy, and real-space resolutions in O 2*p*, O 2*s*, and Ti 3*d* spectra, coupled with scanning tunneling microscopy and density functional theory, constitutes a significant advancement in surface science and catalysis. However, the reliability of this innovative approach necessitates further validation using reference systems, preferably simpler ones with fewer complex coordination environments. The current limitations of the approach are evident in the results and conclusions, highlighting the need for broader applicability. Additionally, several issues must be addressed before considering the publication of this work:

Author reply: We thank the reviewer for finding our study interesting. Further validation of our approach using a simpler reference system is a good suggestion. We made comparison with the results from the simpler rutile-TiO₂(110)-(1×1) surface.

On anatase-TiO₂(001)-(1×4) surface, we observed obvious surface core level shift (SCLS) in the O 2*s* semi-core level spectra (Fig. 3 in the main text). Different from reconstructed anatase-TiO₂(001)-(1×4) surface, the rutile-TiO₂(110)-(1×1) is not reconstructed with only slight atomic relaxation at surface [Charlton *et al.*, *Phys. Rev. Lett.* **78**, 495 (1997)]. Therefore, we compare the O 2*s* spectra obtained from rutile-TiO₂(110)-(1×1) and anatase-TiO₂(001)-(1×4) surfaces with increasing excitation photon energy from 40 eV to 60 eV (Response Fig. 1). We have noted the O 2*s* in the anatase-TiO₂(001)-(1×4) surface contains multiple peaks (Response Fig. 1b). In contrast, the spectrum from rutile-TiO₂(110)-(1×1) surface shows only a peak feature with a broadened range of $E_b = 21\text{-}24$ eV (Response Fig. 1a), similar to the observed single peak feature of O 2*s* in rutile-TiO₂(110) by Jones *et al.* [R. Jones *et al.*, *J. Phys. Chem. C* **126**, 16894 (2022)].

So far, we are aware of few reports about the O 2*s* features in anatase-TiO₂(001)-(1×4). As suggested by the reviewer, this comparison is certainly more helpful to draw attention on the multiple peaks in anatase-TiO₂(001)-(1×4) surface. We thank the

reviewer for the insightful suggestion.

Response Fig. 1. (a) The measured O $2s$ XPS of rutile-TiO₂(110)-(1×1) excited by $h\nu = 41, 45, 50$ and 60 eV, respectively. (b) The corresponding data for anatase-TiO₂(001)-(1×4) excited by $h\nu = 40, 45, 50$ and 55 eV, respectively. Anatase-TiO₂(001)-(1×4) surface shows multiple peaks, while rutile-TiO₂(110)-(1×1) surface is featureless with a broad peak.

As for the Ti $3d$ gap state in the rutile-TiO₂(110)-(1×1) surface, it commonly presents as a gap state at 0.9 eV below Fermi level (E_F), which has been assigned to the surface oxygen defects and/or interstitial Ti atoms [Yim *et al.*, *Phys. Rev. Lett.* **104**, 036806 (2010); Wendt *et al.*, *Science* **320**, 1755-1759 (2008)]. This is in sharp contrast to the two gap states observed in the anatase-TiO₂(001)-(1×4) surface.

In summary, we have done XPS measurements using a simpler and relevant rutile-TiO₂(110)-(1×1) surface as a reference, as suggested by the reviewer. It is found that the coordination induced differences are small, and the electronic structures are less complex in the rutile-TiO₂(110)-(1×1) surface. The comparison between rutile-TiO₂(110)-(1×1) and anatase-TiO₂(001)-(1×4) surfaces does give fruitful insight into the effects of the diverse coordination environments in reconstructed anatase-TiO₂(001)-(1×4) surface, which are demanded to be clarified. Our approach in this manuscript does point to the goal. We have added the data in Supplementary Fig. S5a,b, for comparing O $2s$ spectra from rutile TiO₂(110)-(1×1) and anatase TiO₂(001)-(1×4) surfaces.

1. Inferring the existence of ADM and AOM configurations solely from the STM image is initially imprecise. STM image contrasts can stem from various sources, and

deducing the presence of ADM and AOM solely based on one image is not entirely convincing. While the subsequent sections of the article aim to validate the existence of these structures through spectroscopy, robust experimental evidence remains incomplete.

Author reply: We agree with the reviewer. To address this comment, we have performed *in situ* measurements using integrated STM and non-contact atomic force microscopy (NC-AFM) with the same tip mounted on a qPlus sensor. The results are given below in **Response Fig. 2**. The nonuniform STM features on the ridge have been correspondingly resolved by the AFM, under either constant height or constant force modes. It can be seen that although STM gives a blurry nonuniform contrast at the ridge (Response Fig. 2a, left panel), the high-resolution NC-AFM image show separated spots (Response Fig. 2a, right panel). The spots have two distinct widths and heights as shown in the corresponding line profiles (Response Fig. 2b). Because the AOM involves one more O atom that can increase the repulsive force, we assign the brighter spots to AOM structures and the dimmer one to ADM structures in the AFM image. Therefore, by comparing the same area STM and NC-AFM images, we suggest that the nonuniform STM contrast indeed imply the presence of alternate AOM and ADM structures.

Response Fig. 2. (a) A set of *in-situ* empty-state STM (1.5 V, 10 pA) and the NC-AFM images at a frequency shift of -29 Hz within the same area, measured at 5 K, using a W tip. (b) Line profiles extracted from the corresponding colored lines in (a). The line profile from STM image (red curve) also shows the nonuniform electronic distributions, and the one from AFM (blue curve) present two distinct contrasts, labeled as ADM and AOM, according to the relative height in their models.

We have added these results as new data in Fig. 1d-e and the corresponding description in this revised manuscript (page 6) to illustrate the distinct contrast in qPlus-

AFM image could be evidence for the existence of ADM and AOM configurations, it reads

We perform *in situ* STM and AFM experiments to characterize the structures at a high resolution. The empty-state STM image mainly integrate the Ti electronic state on the ridge and terrace (Fig. 1d, left panel), while the AFM image (given in inverted contrast because of attraction force regime in imaging) mainly reflect the contrasts of O atoms along the ridges (Fig. 1d, right panel). The two groups of O atoms can be distinguished from their relative contrasts, as labeled by ADM and AOM according to the relative heights in their structural models^{32,38} (Fig. 1e). Both of the STM and the AFM images indicate the coexistence of the ADM and AOM structures on the ridge.

2. For the 1:1 mixed ADM-AOM structure, there might be another setup where two AOM patterns are together. Can its energy related to the oxygen chemical potential be added to Figure S6? Also, it would help if the formula used to calculate the phase diagrams for the different ADM-AOM structures is provided.

Author reply: Thanks for the suggestion. We have added this setup where two AOM patterns are together to Supplementary Fig. S9 (also shown below in Response Fig. 3). The previous setup is marked as “1:1-1” and the added one is marked as “1:1-2”. The optimized structures show that the 1:1-1 and 1:1-2 have nearly identical surface energies (difference within ~5 meV), and give almost the same curves in phase diagrams. The corresponding formula to calculate the phase diagrams have been added in the section of Methods. It is noted that similar phase diagrams for AOM and ADM have been calculated in refs. 47,55,56 [*RSC Advances* **12**, 28178-28184 (2022); *Nano Lett.* **16**, 132-137 (2015); *Phys. Chem. Chem. Phys.* **19**, 16615-16620 (2017)]. Our study provides additional information for the mixed ADM-AOM structures with tunable ratios.

Here, we have added the corresponding calculation formula in Methods of the revised manuscript for the calculations of the surface energies and phase diagrams of the mixed ADM-AOM structures.

Response Fig. 3. The stabilities and phase diagrams of the mixed ADM-AOM structures. (a) The optimized structures of pure ADM, 3:1 (the numbers denote the ADM:AOM ratio); 1:1, 1:3 and pure AOM. The surface O atoms of O_{TOP}, O_{SIDE} and O_{TERRACE} are colored by red, brown, and green, respectively. Although the two setups marked as 1:1-1 and 1:1-2 have different structures, they have nearly identical surface energies (difference within ~5 meV). (b) The calculated surface energies of the five structures in (a), plotted as a function of oxygen chemical potential $\Delta\mu_{\text{O}}$ for two different lattice constants of $a = 3.8 \text{ \AA}$ (left panel) and $a = 3.9 \text{ \AA}$ (right panel), respectively. A larger lattice constant $a = 3.9 \text{ \AA}$ is used due to anatase-TiO₂(001) thin films epitaxially grown on 0.7 wt% Nb-doped SrTiO₃(001) substrates. (c) The corresponding phase diagrams with oxygen pressure and annealing temperature, calculated on the basis of the lattice constants of $a = 3.8 \text{ \AA}$ (left panel) and $a = 3.9 \text{ \AA}$ (right panel), respectively.

3. Given the many ways Ti³⁺ can be arranged near the Ov site, it's essential to look at all these configurations to determine the most stable configuration of the defective anatase-TiO₂(001). It's not clear if the authors used spin-polarized calculations during the structural relaxation processes for the defective anatase-TiO₂(001) or if it was only used for single point electronic structure calculations.

Author reply: We thank the reviewer for this suggestion. As it is noticed, ref. 40 [Bigi et al., *Phys. Rev. Mater.* **4**, 025801 (2020), Fig. 6] has systematically studied the possible configurations of Ov defects at anatase-TiO₂(001)-(1×4). They also found the adsorption of O₂ can quench the gap states (GSs), and thus assigned the origin of GSs to the surface Ov defects.

Based on such valuable information, we considered four kinds of possible Ov

configurations at surface, as labeled by O_V-1 (O_V-RIDGE), O_V-2 (O_V-SIDE), O_V-3 (O_V-3C) and O_V-4 (O_V-TERRACE) in Response Fig. 4a. The formation energies of the O_V configurations are calculated by PBE functional with spin-polarized calculations and listed in Response Fig. 4b. It can be seen that the O_V-1 (O_V-RIDGE) is the most likely to appear at ridge sites, and the O_V-3 (O_V-3C) and O_V-4 (O_V-TERRACE) with energy difference of 0.6 eV may co-exist at terrace sites.

In order to calculate the GSs more accurately, we performed the spin-polarized DFT calculation during the structural relaxation processes with PBE+U functional (U = 3.9 eV). The re-calculated DOSs for O_V-1 (O_V-RIDGE), O_V-3 (O_V-3C) and O_V-4 (O_V-TERRACE) sites are shown in Response Fig. 5. The antiferromagnetic state (Response Fig. 5a-c) has a lower formation energy than the ferromagnetic (FM) state (Response Fig. 5d-f). The excess electrons are mainly distributed in adjacent Ti atoms, reducing the Ti⁴⁺ to Ti³⁺, implying the formation of a small electron polaron [*J. Phys. Chem. C* **113**, 14583-14586 (2009)].

Response Fig. 4. Formation energies of possible surface O_V configurations. (a) Side view of the (1 × 4) reconstructed anatase TiO₂(001) slab model. All possible surface O_V configurations are represented with circles. The atoms of O_{TOP}, O_{SIDE}, O_{TERRACE} and O_{BULK} are colored differently. (b) O_V formation energies (eV) at different surface sites. The O_V-1 (O_V-RIDGE) is the most likely to appear at the ridge sites, and the O_V-3 (O_V-3C) and O_V-4 (O_V-TERRACE) with energy difference of 0.6 eV may co-exist at the terrace sites. (c-f) Relevant O_V defect configurations for O_V-1 (O_V-RIDGE), O_V-2 (O_V-SIDE), O_V-3 (O_V-3C) and O_V-4 (O_V-TERRACE), respectively. O_V sites are marked by black arrows.

Response Fig. 5. The calculate the GSs of different O_V sites with antiferromagnetic and ferromagnetic (FM) states. Calculated charge density contours of excess electrons and DOSs of total (gray) and 1-3 layer (yellow: spin up, blue: spin down) based on ADM model with an O_V -1 ($O_{V-RIDGE}$) at ridge (a,d), an O_V -3 (O_{V-3C}) at terrace (b,e) and an O_V -4 ($O_{V-TERRACE}$) at terrace (c,f), respectively. The antiferromagnetic state (a-c) has a lower formation energy than the ferromagnetic state (d-f). DOS of 1-3 layers is extracted to eliminate the effects of (1×1) surface at the last layer, the calculated bandgap is estimated to ~ 2.5 eV. Charge density contours of the excess electron states induced by O_V defects. The excess electrons are mainly distributed in adjacent Ti atoms, reducing the Ti^{4+} to Ti^{3+} , implying the formation of a small electron polaron.

To address comments 3, we have added Response Fig. 4 and Fig. 5 to Supplementary Fig. S11 and Fig. S12, respectively. The antiferromagnetic results in Response Fig. 5 are merged into Figure 4e,f in the revised manuscript. We have added a corresponding discussion in this revised manuscript on pages 14-15, it reads

We calculate the configurations and formation energies for possible surface O_V defects at ridge and terrace sites using spin-polarized DFT (Supplementary Fig. S11). From the

formation energies, it is found that the missing of one O_{TOP} is the most possible O_{V} defect at ridge ($O_{\text{V-RIDGE}}$, Fig. 4e), and the missing of a bridging O atom along [100] direction or along [010] direction could be the O_{V} defect at terrace ($O_{\text{V-3C}}$ and $O_{\text{V-TERRACE}}$, Fig. 4f). The calculated electronic structures with PBE+U functional ($U = 3.9$ eV)⁴⁰ show that each O_{V} defect can contribute to a GS with its excess electron redistributing to adjacent Ti atoms (Fig. 4e-f). The charge redistribution and the distortion of the lattice in the vicinity imply the formation of small electron polaron, similar as the small polaron of O_{V} at rutile- $\text{TiO}_2(110)$ surface⁵⁷. But, from the pDOS with either antiferromagnetic⁵⁸ or ferromagnetic states (Supplementary Fig. S12), the energies of different O_{V} defects are not separated clearly, making the assignment difficult.

4. The authors have given theoretical proof for the presence of both ADM and AOM structures. But is there direct proof for both defective structures being present at the same time? What are the formation energies for these two kinds of oxygen vacancies?

Author reply: (1) After performing the NC-AFM experiments (Response Fig. 2), we think the two distinct contrasts in the qPlus-AFM image could be direct evidence for the presence of alternated ADM and AOM structures in real-space. Also, another evidence could be found in O 2s spectra the Fig. 3f in the main text, in which the ADM ratio is increasing and the AOM ratio is decreasing by controlling the surface reduction. (2) The AOM is the fully oxidized form with one added O atom to the ADM structures in each lattice. Under reduction, the AOM loses one O to ADM, and then loses the second one to form a defective $O_{\text{V-RIDGE}}$. Therefore, the defective $O_{\text{V-RIDGE}}$ structures of AOM and ADM are indeed the same, and it is not expected to observe any difference from them.

5. The study seems to focus on single O_{V} . What happens when there are multiple O_{V} , in terms of shape and electric properties? How do these findings compare to what's seen in experiments?

Author reply: As shown in Response Fig. 4 above, there are three possible kinds of

single O_V defects at [the O_V -2 (O_{V-SIDE}) has the highest energy and is not considered]. Here, to investigate the multiple O_V , we adopt the same kind (Response Fig. 6a-e) or different kinds of O_V (Response Fig. 6f-i) to construct the O_V pairs below:

1) same kind of O_V pairs. Response Fig. 6a-e show the O_V pairs consisting of two neighboring O_V defects. It can be found that the O_V -(1,1) pair at ridge sites has the lowest formation energy of 8.7 eV (Response Fig. 6a). Experimentally, the intensity of GS1 can continue to increase while that of GS2 quickly achieve saturation (Fig. 4d in the main text), suggesting that GS1 related defects can be reasonably assigned to ridge sites. However, it is 0.7 eV higher than the sum of the formation energies of two separated O_V -1. Thus, it suggests such O_V pairs might be quite unstable. This is in line with the previous studies at rutile-TiO₂(110)-(1×1) surface that the O_V pairs are thermodynamically unstable with repulsion to dissociate into separated single O_V 's, as reported by our group [Cui *et al.*, *J. Chem. Phys.* 129, 044703 (2008)] and other group [Zhang *et al.*, *Phys. Rev. Lett.* 99, 126105 (2007)].

2) different kinds of O_V pairs. Response Fig. 6f-i show the O_V pairs consisting of two different kinds of separated single O_V 's. The O_V -(1,3) and O_V -(1,4) pairs have the lower formation energy. We further compare the electronic properties of multiple O_V 's, for example O_V -(1,3,4), as shown in Response Fig. 7. We find that the charge density contours and configurations are basically a direct superposition of separated single O_V features, the GSs show very close energies within the underestimated bandgap of 2.5 eV (Response Fig. 5). The antiferromagnetic state (Response Fig. 7a) has a lower formation energy than the ferromagnetic (FM) state (Response Fig. 7b).

Response Fig. 6. Formation energies of multiple O_V defects. Formation energies of same (a) and different (f) kinds of O_V pairs. (b-e) Relevant same kind of O_V pair configurations for O_V -(1,1), O_V -(2,2), O_V -(3,3) and O_V -(4,4) pairs, respectively. (g-i) Relevant different kinds of O_V pair configurations for O_V -(1,3), O_V -(1,4) and O_V -(3,4) pairs, respectively.

Response Fig. 7. The calculated GSs of multiple O_V-(1,3,4) site with antiferromagnetic and ferromagnetic (FM) states. Calculated charge density contours of excess electrons and DOSs of total (gray) and 1-3 layer (yellow: spin up, blue: spin down) based on ADM model with an O_V-1 (O_V-RIDGE) at ridge, an O_V-3 (O_V-3C) and an O_V-4 (O_V-TERRACE) at terrace. The antiferromagnetic state (a) has a lower formation energy than the ferromagnetic (FM) state (b).

To address this comment 5, we have added the Response Fig. 6 to Supplementary Fig. S13. The antiferromagnetic results in Response Fig. 7 are merged into Figure 4g in the revised manuscript. We have added some sentences in this revised manuscript on page 15, it reads

Such energy inaccuracy is possibly because the self-interaction error in DFT, which usually leads to an underestimated bandgap of TiO₂. In particular, when multiple O_V defects are considered (Supplementary Fig. S13), the GSs show very close energies within the underestimated bandgap of 2.5 eV (Fig. 4g).

Reviewer #2 (Remarks to the Author):

This is an interesting paper, although I do not consider it as a major advance. The authors certainly oversell the work. There is also a major need for extensive correction of the grammar and English which is quite bad in many places.

Author reply: We thank the reviewer for finding our manuscript interesting. TiO₂ seems as a simple binary oxide, but the existing complicated structures and electronic states have not been well understood. In particular, different reconstructed surfaces may appear and play important roles on many physical and chemical processes. Here, our study, using the prototypical surface of anatase-TiO₂(001)-(1×4), demonstrates how the multi-aspect information can be obtained and used to univocally determine the structure-property relationships.

Identification of the surface electronic states is of importance in many surface-related processes of TiO₂. To establish the intricate coordination environment of O atoms with the electronic structures as possible as we can, our approach included the characterization using microscopic and spectroscopic techniques together. Our results may give insights into several aspects, as: (1) the identification of O *2p* surface states from the bulk ones using momentum-resolved band structure from ARPES; (2) identification of the surface states due to existing of ADM and AOM surface structures from the O *2s* spectra; (3) identification of gap states from O_v's at ridge and the terrace sites. These aspects have not been clearly documented before. As an important but intricate reconstructed surface, obtaining such global parameters provides insightful understanding, and represents a major advance. We believe our findings will benefit to a wide range of readers.

We have revised the grammar and English as suggested by this reviewer, and also revised the manuscript substantially following the comments by other reviewers. We appreciate if this reviewer may find our revised manuscript has been improved.

I have a few comments:

Abstract needs work: The sentence “We resolve each...” needs rewriting.

Author reply: We have rewritten the abstract.

The term “multi-domain” seems to be wrong.

Author reply: We've revised the abstract.

Many other parts of the text need checking by a native English speaker, they are somewhat clumsy or have the wrong words. I won't comment further.

Author reply: The English in the revised manuscript has been carefully polished.

Please replace reference 21 by an original reference, it is a recent derivative. The method has been known for 40 years.

Author reply: We've already replaced this reference with a review article [Su *et al.*, *Chem. Rev.* 115, 2818-2882 (2015)].

The O 2s is semicore, not core. Was relaxation or a Slater 1/2 method used? It is standard that positions and even shapes are not correct if just the pDOS is used.

Author reply: Thanks for reminding, we have modified “O 2s core” to “O 2s semi-core”. In our DFT calculation, the core electrons of atoms were considered through the projector augmented wave (PAW) approach. The valence electrons Ti ($3s^23p^63d^24s^2$) and O ($2s^22p^4$) are described by basis sets constituted by plane waves with kinetic energy cutoff of 500 eV. To ensure accuracy of electronic energy, the PBE exchange-correlation functional are employed in all calculation, without any LDA-1/2 or PBE-1/2 method.

Reviewer #3 (Remarks to the Author):

The authors report a multi-technique study (STM, UPS, XPS, ARPES, DFT) to characterize the (4×1) reconstructed TiO₂ anatase (001) surface. This is certainly an interesting and important surface, and new knowledge about it is always a good thing. A model for the (4×1) surface was described by Lazzeri and Selloni (ref. 30) and later a somewhat modified version by Wang et al (ref. 31).

The authors analyze differentiated ARPES valence band spectra and, with the help of DFT calculations and projected DOS, assign them to different O atoms. The authors argue that they are able to distinguish the signals in the valence band for O atoms in 7 different environments. This is a rather naïve view of electronic structure, to the extent that I would view this analysis as an over-interpretation. All bands are hybridized. It makes not much sense to assign them to individual O atoms.

Author reply: We are grateful to the reviewer finding our studied system is certainly an interesting and important surface.

We also thank the reviewer for pointing out some inappropriate states in our previous manuscript, making our analysis somewhat over-interpreted. We have revised our manuscript.

As the reviewer pointed out, all bands from O atoms are hybridized, which contributes to the E - k dispersion. In our study, we have shown that it is still possible to resolve surface O from the bulk even their existing intricate hybridization. The main idea is that the terminating surface O atoms have different coordination environments, which usually contribute to unique electronic states different from the bulk ones.

As the results shown in Fig. 2c (in the main text), the labeled S_1 and S_2 can be assigned to specific surface O atoms. It is seen that the S_1 and S_2 are nearly flat with small momentum range near $\bar{\Gamma}$ point (Fig. 2c in the manuscript). The near flat band feature implies they are localized to specific O sites in real-space. Although the pDOSs indicate the S_1 and S_2 are hybridizing in a broad energy range, the densities of S_1 and S_2 dominantly appear at $E_b \sim 6.8$ eV and 7.2 eV, respectively (Fig. 2d in the manuscript). Plotting the distribution of S_1 and S_2 indicates that they are localizing at O_{TERRACE} and O_{SIDE} (Fig. 2e), respectively. We also carried out the experiment by depositing

potassium (K) to anatase-TiO₂(001)-(1×4). The phenomena of killed S₁ and S₂ by K adsorption further confirm their surface-state nature. We also included the calculated VB dispersion according to the weightings of O_{TOP}, O_{TERRACE}, O_{SIDE} and O_{BULK}, as Supplementary Fig. S3 in the revised version. Here, we also give it below as Response Fig. 8.

Response Fig. 8. Calculated VB dispersion according to the weightings of O_{TOP}, O_{TERRACE}, O_{SIDE} and O_{BULK} based on ADM model. The weightings of surface O atoms (a-c) show local distribution relative to the that of bulk O atoms (d). The weightings of S₁ and S₂ indicate near flat character in the vicinity of $\bar{\Gamma}$ point.

The major revisions include: using a new title “Unveiling diverse coordination-defined electronic structures of reconstructed anatase TiO₂(001)-(1×4) surface”, rewriting the abstract and the introduction, and the analysis about the O-related statements in the whole paper. In the revised manuscript, we have the topic more focused on the resolving of surface states from the different surface coordination environments using the multi-technique approach to draw the much global parameters, by combining with the calculations.

To address this comment, we have added the Response Fig. 8 to Supplementary Fig. S3. We have added some sentences in this revised manuscript on pages 8-9, it reads Furthermore, we plot the band dispersion of each O atoms calculated using the ADM model (Supplementary Fig. S3). The weightings of surface O atoms show localized features relative to the that of bulk O atoms. The weightings of S₁ and S₂ indicate near flat character in the vicinity of $\bar{\Gamma}$ point (Fig. 2f), in good agreement with those

observed in the experimental $E(k_{||})$ map. For O_{TOP} atoms, it is known that they mainly contribute to the VBM^{46,47} with localized features (Fig. 2e). The calculated pDOS of $O_{\text{TOP-ADM}}$ and $O_{\text{TOP-AOM}}$ with ADM and AOM models, respectively, dominate at VBM, but with slight energy difference (Fig. 2d).

The paper then goes on to analyze O2s (shallow core levels), taken at different photon energies, and, again assign them to different O atoms according to PDOS calculations. This makes a little bit more sense, but I am doubtful that the rather pronounced changes in the core levels would only be due to IMFP length effects, as claimed. Then this should lead to a rather smooth change, which should not be very pronounced in the $h\nu$ range considered (39 to 200 eV).

The observed relatively drastic intensity changes could be due to two phenomena: 1) hybridization with Ti band (and the well-known resonant photoemission effect) and/or 2) if taken with the energy analyzer set to a narrow angular acceptance range, photoelectron diffraction.

As far as I can see, neither of these effects has been considered.

Author reply: Thank the reviewer finding the analysis O 2s more sense. As far as we understand, this is for the first time the experiment using O 2s semi-core levels to analyze the intricate electronic states of the reconstructed surfaces.

We also thank the reviewer for suggesting other possible phenomena. The comments are addressed below.

- (1) As stated by the reviewer, “*Then this should lead to a rather smooth change, which should not be very pronounced in the $h\nu$ range considered (39 to 200 eV).*”, he or she thought that the O 2s may not have a pronounced intensity change in the small $h\nu$ range. After comparing the O 2s spectra between rutile-TiO₂(110)-(1×1) and anatase-TiO₂(001)-(1×4) surfaces (Response Fig. 9), one can see that the O 2s at anatase-TiO₂(001)-(1×4) contains much fruitful information with multiple peaks and pronounced intensity changes (Response Fig. 9b), instead of the single peak feature of the simpler surface structure of rutile-TiO₂(110)-(1×1). Such a difference may reflect the different contributions

from the diverse coordinated O atoms in the reconstructed anatase $\text{TiO}_2(001)$ - (1×4) surface.

Response Fig. 9. O 2s spectra and resonant photoemission process. (a) The measured O 2s XPS of rutile- $\text{TiO}_2(110)$ - (1×1) excited by $h\nu = 41, 45, 50$ and 60 eV, respectively. (b) The corresponding data for anatase- $\text{TiO}_2(001)$ - (1×4) excited by $h\nu = 40, 45, 50$ and 55 eV, respectively. Anatase- $\text{TiO}_2(001)$ - (1×4) surface shows multiple peaks, while rutile- $\text{TiO}_2(110)$ - (1×1) surface is featureless with a broad peak. (c) Evolution of O 2s XPS by varying the photon energy from 39 eV to 55 eV in 1 eV step. The strength of bulk increases with the increase of the excitation energy (gray line). But, resonant photoemission process occurs at around $h\nu \sim 43$ eV for surface level (red arrow) and $h\nu \sim 46$ eV for bulk level (black arrow). All measurements were performed at 20 K.

(2) We further consider the resonant photoemission effect. It has been shown that a resonance process can occur at around $h\nu \sim 47$ eV for Ti $3p \rightarrow 3d$ optical transition at anatase- $\text{TiO}_2(001)$ surface (refs. 53,54) [Thomas et al., *PRB* 67, 035110 (2003); *PRB* 75, 035105 (2007)]. It is seen that the bulk peak at 22.7 eV is monotonously increasing from low-to-high $h\nu$, and becomes dominant at $h\nu > 120$ eV (Fig. 3a in the main text), indicating the bulk peak intensity is mostly related to the electron inelastic mean free path (IMFP). Here, we have done more experiments using variable $h\nu$ (39 - 55 eV) excitations. The $h\nu$ -dependent data are given below in Response Fig. 9c (also added in Supplementary Fig. S5). It is seen that the surface peaks (in the range of about 18 - 22 eV) show a maximum

intensity at about $h\nu \sim 43$ eV (red arrow), and a local maximum intensity at about $h\nu \sim 46$ eV (black arrow) for the bulk one. The appearance of the maximum intensity could be from the resonance by hybridization with Ti band. Even though, the obvious different behaviors from the different peaks (as the labeled bulk and surface states) do indicate their different origins, which may further rationalize our analysis.

Therefore, we can see that we may still separate the surface states from the bulk ones, even the resonant photoemission effect may occur.

(3) As for the possible effect of photoelectron diffraction, we can exclude this possibility in our experiment. The reviewer mainly concerned the possible enhancements of the ARPES signals at certain angles, because of the diffraction of photoelectrons by the surface atoms (lattice). Considering the excitation photon energy of $h\nu = 40\text{-}55\text{eV}$, the kinetic energy (E_{kin}) of the outgoing photoelectrons at (001) surface is about 12-27 eV, which falls in the range of those O $2s$ signals that we discussed. In this kinetic energy range, the photoelectron wavelength is in the range of $\lambda = 2.36 \sim 3.54$ Å. Using $2d\sin\theta = \lambda$, we get the estimated angles $\theta = 19\sim 28^\circ$ for the first-order diffraction peak using the unconstructed lattice constant of 3.80 Å. These angles are obviously larger than the incident angle limit of 15° of the slit used in the ARPES. However, this may not totally exclude the possible diffraction from the (1×4) reconstructed surface structure.

As a further confirmation, we made additional analyses using the angle distribution curves (ADCs) and the energy distribution curves (EDCs). As shown in Response Fig. 10, such analyses provide more evidence to exclude the possibility that the photoelectron diffraction may affect the O $2s$ spectra by the reconstructed surface structure. Response Fig. 10a-c show the ARPES spectra measured at the excitation photon energies of 39, 43 and 46 eV, respectively. The ADCs, obtained correspondingly by integrating the signals within the E_b

range of 18~24 eV of each ARPES spectrum, are superimposed on Response Fig. 10a-c. It is seen that the ADCs at each photon energy give overall Gaussian-like shape for the integrated intensity distributions against θ , obviously no diffraction-enhanced intensity with $\pm 15^\circ$. Moreover, Response Fig. 10d-f show the EDCs, which were obtained by cutting at the given angles (in each curve, the signals were integrated within $\pm 1^\circ$ at the labeled angle), in comparison with the normalized total signals (integrated signals with the whole range of $\pm 15^\circ$). It is obvious that the EDCs show nearly the same feature at each excitation photon energy, showing the angle-independent O 2s spectra. These analyses can exclude the possibility of the diffraction effect by the constructed surface structure. Therefore, on the basis of the distinguishable features from those of bulk ones, our observations of the multiple peaks of the O 2s spectra can be assigned to the diverse coordination environments of the surface atoms.

Response Fig. 10. Angle - and energy-dependent O 2s spectra. (a-c) Three selected E_b - θ cuts with $h\nu = 39, 43$ and 46 eV excitation. The angle distribution curves (ADCs) are marked by white

lines (signal integration within the E_b range of 18~24 eV). (d-f) The corresponding energy resolution curves (EDCs) at different scattering angles (in each curve, the signals were integrated within $\pm 1^\circ$ at the labeled angle), in comparison with the normalized total signals (integrated signals with the whole range of $\pm 15^\circ$). It is obvious that the EDCs show nearly the same feature at each excitation photon energy, showing the angle-independent O 2s spectra.

To address this comment, we have added the Response Fig. 9 to Supplementary Fig. S5, the Response Fig. 10 to Supplementary Fig. S7 and added this discussion in the revised manuscript on page 11, it reads

We can exclude the effect of the photoelectron diffraction due to the possible diffraction-caused enhanced angle-dependent intensity variations (Supplementary Fig. S7). The angle distribution curves (ADCs) at each photon energy give overall Gaussian-like shape for the integrated intensity distributions against θ , obviously no diffraction-enhanced intensity with $\pm 15^\circ$. The EDCs show nearly the same feature at each excitation photon energy, showing the angle-independent O 2s spectra. These analyses can exclude the possibility of the diffraction effect by the reconstructed surface structure. While, by measuring the spectra using the tunable excitation photon energy in the range of $h\nu \sim 39 - 55$ eV, the resonant photoemission processes were observed to occur at around $h\nu \sim 43 - 46$ eV (Supplementary Fig. S5c), which could be assigned to the Ti $3p \rightarrow 3d$ optical transition at anatase-TiO₂(001) surface^{53,54}. Such resonant photoemission processes could enhance the photoemission intensities of the surface semi-core levels at certain excitation photon energy, and make the peaks more distinguishable in the O 2s spectra. Nevertheless, this effect does not obviously contribute any additional peak and much easily be recognized according to the analysis of our results.

The analysis of the Ti3p core level spectra and the assignment of the defect state to different O vacancies makes sense only if the excess electrons are in a small polaron state right next to the vacancy. This is possibly true, but more work would be needed to show this.

Author reply: The referee is correct that small polarons are formed at the nearby Ti atoms from O vacancies. In the Ti $3d$ spectra, there are actually two kinds of defect states formed by the excess electrons: the gap states (GS1 and GS2) and metallic state (MS) at E_F . The MS is delocalized electron gas, as has been revealed as large polarons in our (refs.39) [*Nano Lett.* **21**, 430-436 (2021)] and other groups' previous studies [Moser *et al.*, *Phys. Rev. Lett.* **110**, 196403 (2013)]. The GS1 and GS2 are the main focus in this manuscript, whose origins are assigned to ridge ($O_{V-RIDGE}$) and terrace (O_{V-3C} and $O_{V-TERRACE}$) sites, respectively.

Charge density contours of the excess electron states induced by $O_{V-RIDGE}$, O_{V-3C} and $O_{V-TERRACE}$ are plotted in Supplementary Fig. S12 (also shown in Response Fig. 5 above), which show the excess electrons are mainly distributed in adjacent Ti atoms, reducing the Ti^{4+} to Ti^{3+} . Meanwhile, local distortions of the lattice take place in the vicinity (Ti sites), implying the formation of a small electron polaron [*J. Phys. Chem. C* **113**, 14583-14586 (2009)]. This understanding is in line with the small polarons of O_v at rutile- $TiO_2(110)$ surface (ref. 58) [Setvin *et al.*, *Phys. Rev. Lett.* **113**, 086402 (2014)].

Summarizing, the interpretation of the experimental data is naïve at best, and completely wrong at worst.

Author reply: With full respect, we cannot totally agree with the reviewer, although we can accept his/her technically good suggestions. Accordingly, we have made revision according to the reviewers' comments. It's our sincerely hope that the reviewer may find the manuscript has been substantially improved.

Reviewer #4 (Remarks to the Author):

The paper by Ma *et al.* reports on the assignment of different coordination structures for anatase TiO₂(001) with their corresponding electronic structures. The work is a multi-technique experimental approach combined with DFT calculations. While STM images provide information about the coexistence of two specific coordination environments at the ridge sites, with ARPES and XPS, the authors get insights into the electronic structure. However, it is the combination of these experimental techniques with theoretical DFT calculations that allows the determination of seven local coordination environments in the anatase TiO₂(001)-(1×4) surface reconstruction. The subject of this research is of great interest due to the relevance of this material, TiO₂, in different technological applications, being of special importance in catalysis due to its surface reactivity. The manuscript achieves a high enough scientific ranking to be accepted in Nature Communication. The authors' experiments are well done and well thought out and theoretical calculations are fundamental in assigning the different electronic states to the coordination environments. The work not only demonstrates the different coordination structures of the anatase TiO₂(001)-1×4 surface, but also provides a paradigm to explore the structure and electronic properties of TMOs. However, there are some minor points that the authors need to address:

Author reply: We thank the reviewer for the insightful comments and finding our results interesting. We have addressed all the comments and revised the manuscript accordingly.

- Catalysis and photocatalysis are not intrinsic properties of materials per se, but rather phenomena that arise from the interaction between materials and chemical reactions (and light in the case of photocatalysis). Therefore, it is inaccurate for the author to state that TMOs "exhibit versatile functional properties such as catalysis and photocatalysis" being more appropriate that TMOs possess versatile functional properties suitable for various applications, including catalysis and photocatalysis. Similarly, strong correlation is not a property but a phenomenon. The sentence "These fascinating properties ranging from strong correlation to surface catalysis ..." is not entirely correct.

More precisely, the authors may refer to “These fascinating phenomena ranging from strong correlation to surface catalysis ...”.

Author reply: We thanks the reviewer for the kind correction of the descriptions. We have revised all these descriptions.

- In page 7 and 8, “OTOP-ADM and OTOP-ADM” is written several times instead of “OTOP-ADM and OTOP-AOM”.

Author reply: Thanks for point out these typos. We have revised.

- In the case of the O-2s spectra fits (Fig. 3b-c), are the minimum four sub-spectra needed to match the XPS O-2s curve? Would it be possible to achieve a good fit with fewer curves? Clarify in the manuscript.

Author reply: We agree that the multi-peak fitting in XPS spectrum is tricky. We did this carefully by taking account of: first, the kinks and shoulders in the spectrum could roughly point to the possible peaks, as shown by the green arrows in the raw spectra and the corresponding spectra after background subtraction (Response Fig. 11a,b); second, in a series of spectra with different excitation $h\nu$, the corresponding peaks should keep at the same energy, but change in intensity, as labeled by the dashed lines (Response Fig. 11c).

As suggested, we also tried a three-peak fit for the same spectra in Response Fig. 11d. It can be found that the middle peak (blue shade) cannot well reproduce the two kinks at 21.2 and 20.1 eV, and the energies of middle (blue shade) and right (brown shade) peaks are changing at different $h\nu$. Thus, the three-peak fit is apparently worse than the four-peak fit.

To address this comment, we have added the Response Fig. 11 to Supplementary Fig. S6 with these fitting details.

Response Fig. 11. Peak fitting of O 2s spectra. (a) The measured O 2s XPS of anatase-TiO₂(001)-(1×4) excited by $h\nu = 39, 43$ and 46 eV, respectively. The backgrounds (BGs) to be subtracted are shown with grey lines. (b) The corresponding spectra after BG subtraction. The green arrows point to possible peaks. Peak fitting of O 2s spectra for four-peak fit (c) and three-peak fit (d), respectively. (c) are same as Fig. 3b-d in the main text.

- The metallic state in the XPS Ti-3d spectra (Fig. 4) appears already after 1 min of light irradiation and remains almost constant for longer. Comment on it.

Author reply: Thanks for the good question. This is because the metallic state (MS) is not a single peak, but an electron pocket evolving below E_F . The MS is formed by the excess electron doping from O_v defects, which causes the band bending and thus drags the conduction band minimum below E_F to form an electron pocket. This had been investigated in our previous study (ref. 39) [*Nano Lett.* **21**, 430-436 (2021)] and other group's study (ref. 40) [Bigi *et al.*, *Phys. Rev. Mater.* **4**, 025801 (2020)]. With the high flux synchrotron light, 1 min irradiation can create enough O_v defects to induce both the GSs and MS. With longer irradiation, the intensities of GSs increase gradually, while the intensity of MS looks almost constant.

We use our previous data (Fig. S3 in ref. 39) to illustrate how the electron pocket of MS evolves under longer irradiation (Response Fig. 12). To observe an intact electron pocket, the data should be collected in the second Brillouin zone Γ_{10} , because the matrix element effect causes the vanishing intensity in the first Brillouin zone Γ_{00} . As shown in Response Fig. 12, with longer time irradiation, the intensity at E_F remains almost unchanged, but the electron pocket (indicated by the white dashed curves) becomes larger with an increasing Fermi wavevector k_F . The increasing electron density n under longer irradiation can be obtained by $n = k_F^3/3\pi^2$, as shown by the red numbers in Response Fig. 12a.

Response Fig. 12 (adopted from ref. 39). Spectral evolution of the raw ARPES cuts (a0-e0), second derivative cuts (a1-e1) and EDCs at Γ_{10} point (a2-e2) with increasing carrier doping at anatase $\text{TiO}_2(001)-(1 \times 4)$ surface by synchrotron irradiation. Synchrotron irradiation time: 2 min (a), 4 min (b), 8 min (c), 12 min (d) and 15 min (e). The white and yellow arrows point to the plasmonic polaron (PL) and kink structure, respectively.

- In this work, the assignment of the two gap states in the Ti-3d XPS spectra, which were previously observed but not understood until now, is quite important. However,

the Ov-ridge GS has two peaks and, although they are centered at 1.1 eV, one is located at 1.6 eV. This BE value is the same as the GS2 contribution. Could it interfere with the GS assignment? Add some discussion in the manuscript to clarify this point.

Author reply: We thank the reviewer for pointing out such unclarity in our manuscript. As also suggested by the reviewer 1, we have systematically recalculated the possible configurations and electronic structures of surface Ov defects. Four kinds of surface Ov defects, as labeled by Ov-1 (Ov-RIDGE), Ov-2 (Ov-SIDE), Ov-3 (Ov-3C) and Ov-4 (Ov-TERRACE) in Response Fig. 4a, are considered. From their formation energies, it can be found the Ov-1 is the most likely to appear at the ridge sites, and the Ov-3 (Ov-3C) and Ov-4 (Ov-TERRACE) with energy difference of 0.6 eV may co-exist at the terrace sites.

We next re-calculate the pDOS of Ov-1 (Ov-RIDGE), Ov-3 (Ov-3C) and Ov-4 (Ov-TERRACE), and extract their energy and space distributions by using spin-polarized DFT calculation with PBE+U functional ($U = 3.9$ eV). Due to the self-interaction error, DFT usually underestimates the bandgap of TiO₂, which further makes the energies of GSs inaccurate. In our case, the calculated bandgap of anatase-TiO₂(001)-(1×4) is estimated to ~ 2.5 eV, with GSs of Ov-RIDGE, Ov-3C and Ov-TERRACE locating inside the bandgap. In previous version of this manuscript, the calculated GS involves two peaks, which are due to the ferromagnetic states. We have noted the ferromagnetic states are not sufficient to describe the GS of Ov, because each Ov denotes two excess electrons to reduce two Ti⁴⁺ to Ti³⁺, where the two Ti³⁺ undergo antiferromagnetic coupling to a more stable state (ref. 59) [Yang *et al.*, *Phys. Rev. B* **81**, 033202 (2010)]. Now, the antiferromagnetic states are considered in the calculation, which are energetically more favorable and present only same peak of spin for each GS (Response Fig. 13). However, it is found that the GSs of Ov-RIDGE, Ov-3C and Ov-TERRACE locate very close in energy, making it difficult to assign them to the experimentally observed GS1 at $E_b \sim 1.1$ eV and GS2 at $E_b \sim 1.8$ eV directly.

Therefore, we try to assign the GS1 and GS2 more reasonably by taking account of following experimental aspects:

- 1) Bigi *et al.* (ref. 40) [*Phys. Rev. Mater.* **4**, 025801 (2020)] designed a nice experiment to deposit O₂ to quench both the GS1 and GS2, and thus demonstrate their

origin must be from surface, rather than from bulk. This informative experiment enables us to consider the origin of GS1 and GS2 only from surface defects of O_V -1, O_V -3 and O_V -4.

2) Our experiments reveal the intensities of GS1 and GS2 raise asynchronously under increasing light irradiation (Fig. 4d in main text), implying the GS1 and GS2 must have different origins, rather than a same origin.

3) Our previous study detected a gap state at $E_b \sim 0.9$ eV for the defects at ridge sites by scanning tunneling spectroscopy (ref. 32) [*Nat. Commun.* **4**, 2214 (2013)]. This is very close to the GS1 at $E_b \sim 1.1$ eV, making us to consider the origin of GS1 as the O_V -RIDGE from the ridges. Moreover, we have reported another gap state at $E_b \sim 1.6$ eV for the hydroxyl groups at terrace sites (ref. 49) [*J. Am. Chem. Soc.* **144**, 13565 (2022)]. This is very close to the GS2 at $E_b \sim 1.8$ eV. Because the hydroxyl groups provide the same nature as oxygen vacancies to induce excess electrons to form gap states [Valentin *et al.*, *PRL* 97, 166803 (2006)], we suggest the GS2 could probably origin from the terrace defects of O_V -3C and O_V -TERRACE. The O_V -3C and O_V -TERRACE have very similar electronic structures, which are not expected to be distinguished further in experiments.

4) To verify the assignments of GS1 at $E_b \sim 1.1$ eV to O_V -RIDGE, and GS2 at $E_b \sim 1.8$ eV to O_V -3C and O_V -TERRACE, we design an experiment of CH_3OH dissociation at different temperatures to mimic the electron donation at ridge and terrace sites respectively. Our previous temperature programmed desorption (TPD) spectra demonstrated the CH_3OH only adsorbs at ridge sites at $T > 190$ K, but adsorbs at both ridge and terrace sites at $T < 190$ K, which is similar as the temperature dependent adsorption of H_2O at anatase- $TiO_2(001)-(1 \times 4)$ surface (ref. 49) [*J. Am. Chem. Soc.* **144**, 13565 (2022)]. Therefore, at $T > 190$ K, GS1 is expected to appear for ridge sites after CH_3OH dissociation, while at $T < 190$ K, GS1 and GS2 for ridge sites and terrace sites are expected. Response Fig. 14a shows the CH_3OH dissociation induced gap states at different temperatures. At $T > 190$ K, the gap state peak lies at $E_b \sim 1.2$ eV constantly. However, at $T < 190$ K the gap state peak is broadened and shifting to higher energies. The best fit of these peaks can well distinguish two peaks at about $E_b \sim 1.2$ eV and 1.75 eV (Response Fig. 14b), which can be well assigned to the hydroxyl groups induced

gap states at ridge and terrace sites, respectively. As mentioned above, because the gap states formed by hydroxyl groups and oxygen vacancies have the same nature by excess electron reduction of Ti^{4+} to Ti^{3+} , we believe the CH_3OH experiment can well support the assignments of GS1 and GS2 to ridge ($O_{V-RIDGE}$) and terrace (O_{V-3C} and $O_{V-TERRACE}$) sites, respectively.

Response Fig. 13. The calculate the GSs of different O_V sites with antiferromagnetic and ferromagnetic state. Calculated charge density contours of excess electrons and DOSs of total (gray) and 1-3 layer (yellow: spin up, blue: spin down) based on ADM model with an O_{V-1} ($O_{V-RIDGE}$) at ridge (a,d), an O_{V-3} (O_{V-3C}) at terrace (b,e) and an O_{V-4} ($O_{V-TERRACE}$) at terrace (c,f), respectively. The antiferromagnetic state (a-c) has a lower formation energy than the ferromagnetic state (d-f). DOS of 1-3 layers is extracted to eliminate the effects of (1×1) surface at the last layer, the calculated bandgap is estimated to ~ 2.5 eV. Charge density contours of the excess electron states induced by O_V defects. The excess electrons are mainly distributed in adjacent Ti atoms, reducing the Ti^{4+} to Ti^{3+} , implying the formation of a small electron polaron.

Response Fig. 14. (for reviewer only) (a) Evolution of GSs measured by temperature dependent UPS spectra from 240 to 140K with an excess amount of CH₃OH ($P(\text{CH}_3\text{OH}) = 1 \times 10^{-9}$ mbar for 150 seconds) at each temperature. (b) The UPS spectra are fitted after background subtraction with one peak at 1.20 eV at $T > 190$ K, and double peaks at 1.20 and 1.75 eV at $T < 190$ K.

To address this comment, we have added the Response Fig. 13 to Supplementary Fig. S12. The antiferromagnetic results in Response Fig. 13 are merged into Figure 4e,f in the revised manuscript. Since we are preparing another new manuscript, we extracted two temperature data (220 and 160 K) in Response Fig. 14 and merged them into Figure 4h,i in the revised manuscript. We have added some discussion in the revised manuscript on pages 14-15, it reads

But, from the pDOS with either antiferromagnetic⁵⁹ or ferromagnetic states (Supplementary Fig. S12), the energies of different O_V defects are not separated clearly, making the assignment difficult. Such energy inaccuracy is possibly because the self-interaction error in DFT, which usually leads to an underestimated bandgap of TiO₂. In particular, when multiple O_V defects are considered (Supplementary Fig. S13), the GSs show very close energies within the underestimated bandgap of 2.5 eV (Fig. 4g).

Further verifying the assignment of GS1 and GS2 requires to create O_V defects at ridge or terrace selectively at the surface, which is quite difficult in experiments. Alternatively, we design an experiment to create hydroxyl groups at ridge and terrace sites from methanol (CH₃OH) dissociation, by considering the hydroxyl groups can provide the same nature as the O_V defects to induce excess electrons to form GSs. Methanol has similar adsorption behavior as water⁴⁹, that the molecular CH₃OH only adsorbs at ridge

sites at $T > 190$ K, but adsorbs at both ridge and terrace sites at $T < 190$ K. Therefore, at $T > 190$ K, a single defect state is expected to appear for ridge sites after CH_3OH dissociation, while at $T < 190$ K, double defect states for ridge and terrace sites are expected. Figure 4h compare the spectra obtained after CH_3OH dissociation with samples prepared at 160 K and 220 K, respectively. Clearly, for the spectrum at 160 K, only one GS peak at ~ 1.20 eV can be detected; for the spectrum at 220 K, the peak is broadened to higher energies. The best fit can distinguish two peaks at about $E_b \sim 1.20$ eV and 1.75 eV (Fig. 4i), which could be assigned to the hydroxyl groups induced GSs at ridge and terrace sites, respectively. Such similarity strongly suggests that the excess electron doping by either O_v defects or hydroxyl groups can form a GS at $E_b \sim 1.1-1.3$ eV at ridge and a GS at $E_b \sim 1.6-1.8$ eV at terrace.

- Several sentences in the manuscript suffer from clarity and grammar issues. The manuscript requires improvement to achieve linguistic correction.

Author reply: Thanks again for pointing out the grammar issues. We have revised many sentences, and hope the reviewer may find our manuscript substantially improved.

REVIEWERS' COMMENTS

Reviewer #1 (Remarks to the Author):

The authors have reasonably addressed all the issues raised in the initial report and have successfully implemented the required changes. Considering these revisions, I suggest the publication of the manuscript in Nature Communications after suitable revisions. The only remaining concern is the distribution of Ti^{3+} ions. It should be noted that for a given V_o configuration, the Ti^{3+} ions can exhibit multiple distributions.

Reviewer #2 (Remarks to the Author):

I have not changed my opinion that this article is not a major advance, and as such does not belong in Nature Communications. It has some good science, albeit not everyone (the authors and reviewers) agrees about all it it), but there is nothing fundamentally new. Reduced oxides have been known for decades, as has different chemical environments. While the authors may want to "sell" this due to the importance of TiO_2 , I do not accept that as at all relevant to publication.

Reviewer #3 (Remarks to the Author):

Initially, I was pretty critical of this work. I am much happier with the revised version: The authors have provided substantial new results from experiments and have used additional techniques, particularly ncAFM. Some of their experiments (e.g., the photon-dependent O_2s core level measurements) have excluded the experimental artifacts I had suspected.

The authors are also more careful with their interpretation, toning down some initial claims.

I feel that the paper is ready to be published.

Reviewer #4 (Remarks to the Author):

I have checked the revised version and am fully satisfied with the changes made by the authors. I believe the paper should be accepted for publication without further modification.

REVIEWERS' COMMENTS

Reviewer #1 (Remarks to the Author):

The authors have reasonably addressed all the issues raised in the initial report and have successfully implemented the required changes. Considering these revisions, I suggest the publication of the manuscript in Nature Communications after suitable revisions. The only remaining concern is the distribution of Ti^{3+} ions. It should be noted that for a given V_o configuration, the Ti^{3+} ions can exhibit multiple distributions.

Author reply: We thank the reviewer for recommending the publication of our manuscript. The reviewer asks about the distributions of the Ti^{3+} state for a given V_o configuration on anatase- $\text{TiO}_2(001)-(1\times 4)$ surface.

We note on rutile- $\text{TiO}_2(110)$ surface, it has been known that the defect charge created by O_v defect is dynamically shared by several subsurface and surface Ti sites, with a dominant contribution from particular second-layer subsurface sites, by using ab initio molecular dynamics (AIMD) simulations [*Phys. Rev. Lett.* 105, 146405 (2010)]. In experiments, the charge is found to delocalize from O_v sites by several lattice distance at 80 K [*J. Chem. Phys.* 130, 124502 (2009)], and condense into small polaron states at <16 K [*Phys. Rev. Lett.* 117, 116402 (2016)].

On anatase- $\text{TiO}_2(001)-(1\times 4)$ surface, our previous STM experiments have indicated the gap state at ridge is very localized to the point defect site [*Nat. Commun.* 4, 2214 (2013)]. Here, to address this comment, we perform AIMD simulations to investigate the localization of excess electrons in three kinds of defective structures of O_v -RIDGE, O_v -3C and O_v -TERRACE (Response Fig. 1). The initial charge distributions at $t = 0$ are shown in Response Fig. 1a-c, with the labeled Ti atom. The excess charge initially localized in adjacent Ti atoms. We performed the PBE + U simulations at $T = 300$ K (Response Fig. 1d-f). The dynamical behavior can be cast into distribution functions using localization histograms, as shown in Response Fig. 1g-i. Clearly, different from the delocalization of excess charge on rutile $\text{TiO}_2(110)$, the results show the excess charge on anatase- $\text{TiO}_2(001)-(1\times 4)$ is highly localized, and eventually settles down to the two nearest Ti atoms (> 95%).

To address this comment, we added a sentence in the main text “Further ab initio

molecular dynamics (AIMD) simulations indicate the excess electrons are stable at the adjacent Ti atoms (Supplementary Fig. 12)”; and added the AIMD simulation results in Supplementary Fig. 12.

Response Fig. 1 (added as Supplementary Fig. 12). The dynamical behavior of excess charge in three kinds of defective structures of O_V-RIDGE, O_V-3C and O_V-TERRACE. (a-c) The initial charge distributions at $t = 0$ with the labeled Ti atom in three O_V defect configurations for O_V-RIDGE, O_V-3C and O_V-TERRACE, respectively. (d-f) Dynamics of the fractional occupation of particular Ti 3d orbitals during a time fragment at $T = 300$ K; At $t = 0$ the charge is localized in the two nearest Ti atoms; populations of about 1 and 0 correspond to Ti³⁺ and Ti⁴⁺ charge states, respectively. (g-i) Distribution function of the average population of available Ti atoms by the two excess electrons obtained from the full simulation (d-f). The excess charge is found to be extremely localized and to eventually visit the two nearest Ti atoms (> 95%) on anatase TiO₂(001).

Reviewer #2 (Remarks to the Author):

I have not changed my opinion that this article is not a major advance, and as such does not belong in Nature Communications. It has some good science, albeit not everyone (the authors and reviewers) agrees about all it it), but there is nothing fundamentally new. Reduced oxides have been known for decades, as has different chemical environments. While the authors may want to "sell" this due to the importance of TiO₂, I do not accept that as at all relevant to publication.

Author reply: We are grateful to the Reviewer for reviewing our revised manuscript again, but we regret that he/she did not change his/her opinion.

While as the Reviewer pointed out “*Reduced oxides have been known for decades*”, he/she may also note that there are many fundamental issues far from well understood in the important system of TiO₂. As he/she stated that “*It has some good science*”, we believe our paper has addressed the fundamentally important characteristic electronic structures from diverse coordination environments on the prototypical anatase-TiO₂(001), which has never been clarified before. Due to the existence of surface reconstruction, such a complicated system has led to some long-term controversies on its surface properties. As an example, one existing debated problem is whether the ridge or terrace sites of anatase-TiO₂(001)-(1×4) are responsible for water splitting [*Phys. Rev. Lett.* 121, 206003 (2018); *J. Am. Chem. Soc.* **144**, 13565 (2022)], although the photocatalytic water splitting by TiO₂ has been known for more than 50 years [Nature 238, 37 (1972)]. To settle this argument, an insight into the intricate coordination environments and surface electronic structures of anatase-TiO₂(001)-(1×4) surface is of course urgently required.,

To this end, our study provides a univocal determination of the characteristic electronic structures from diverse coordination environments by using combined experimental techniques. The revealed coordination (structure)-electronic (property) relationships could pave the way to understand the widely-concerned surface catalysis and correlation phenomena of anatase-TiO₂. Specifically, the main focus in photocatalytic water splitting on TiO₂ is to understand the energy level alignment between the TiO₂ valence bands (VBs) and the highest occupied molecular orbitals

(HOMOs) of water at their interface. Distinguishing the surface VBs of O_{TOP} , O_{TERRACE} and O_{SIDE} can directly locate and determine how the water HOMOs can hybridize with the TiO_2 VBs with energy-momentum-site specificities; regarding the correlation phenomena, the anatase- TiO_2 can provide a platform to tune the electron-phonon and electron-plasmon couplings through the excess charge by O_{V} defect [*Phys. Rev. B* **97**, 165113 (2018)] and photo-excitation [*Chin. J. Chem. Phys.* **35**, 270-280 (2022)], and can support an enhanced electron-phonon coupling interface of FeSe/anatase- TiO_2 to induce high- T_{c} superconductivity [*Phys. Rev. Lett.* **117**, 067001 (2016)]. Distinguishing the origins of the doped electrons from different GSs can provide prerequisites to tune the electron-boson couplings in the complicate many-body interactions.

To emphasize the importance of our study, we have added a comment sentence “which requires an insight into the intricate coordination environments and surface electronic structures”, and several related references, as “regarding the correlation phenomena, the anatase- TiO_2 can provide a platform to tune the electron-phonon and electron-plasmon couplings through the excess charge by O_{V} defect^{39,61} and photo-excitation^{62,63}, and can support an enhanced electron-phonon coupling interface of FeSe/anatase- TiO_2 to induce high- T_{c} superconductivity⁶⁴” in the Discussion section. We hope our study can benefit the TiO_2 community, as well as the broad oxides, surface science, and material science communities.

Reviewer #3 (Remarks to the Author):

Initially, I was pretty critical of this work. I am much happier with the revised version: The authors have provided substantial new results from experiments and have used additional techniques, particularly ncAFM. Some of their experiments (e.g., the photon-dependent O2s core level measurements) have excluded the experimental artifacts I had suspected.

The authors are also more careful with their interpretation, toning down some initial claims.

I feel that the paper is ready to be published.

Author reply: We appreciate the Reviewer's initially seemingly critical but actually very insightful comments, which have greatly helped us to improve our manuscript. We are pleased to see that the Reviewer is happier with the revised version, and thank the reviewer for finding the paper is ready to be published.

Reviewer #4 (Remarks to the Author):

I have checked the revised version and am fully satisfied with the changes made by the authors. I believe the paper should be accepted for publication without further modification.

Author reply: We thank the reviewer for appreciation of our revisions and recommending publication of our manuscript.